# Real Time Detection of Airborne Fluorescent Bioparticles in Antarctica

Ian Crawford[1], Martin W. Gallagher[1], Keith N. Bower[1], Thomas W. Choularton[1], Michael J. Flynn[1], Simon Ruske[1], Constantino Listowski[2,*], Neil Brough[2], Thomas Lachlan-Cope[2], Zoë L. Fleming[3], Virginia E. Foot[4], and Warren R. Stanley[5]

[1]Centre for Atmospheric Science, SEES, University of Manchester, Manchester, UK
[2]British Antarctic Survey, NERC, High Cross, Madingley Rd, Cambridge, UK
[3]University of Leicester, Department of Chemistry , Leicester, UK
[4]Defence Science & Technology Laboratory, Porton Down, Salisbury, Wilts, U.K
[5]Science and Technology Research Institute, University of Hertfordshire, UK
[*]now at: LATMOS/IPSL, UVSQ Universitris-Saclay, UPMC Univ. Paris 06, CNRS, Guyancourt, France

*Correspondence to:* I. Crawford (i.crawford@manchester.ac.uk)

**Abstract.** We demonstrate for the first time, continuous real-time observations of airborne bio-fluorescent aerosols recorded at the British Antarctic Survey's Halley VI Research Station, located on the Brunt ice shelf close to the Weddell Sea coast (Lat. 75°34'59"S, Long. 26°10'0"W) during Antarctic Summer, 2015. As part of the NERC MAC (Microphysics of Antarctic Clouds) aircraft aerosol cloud interaction project, observations with a real-time Ultraviolet Light Induced Fluorescence (UV-LIF) spectrometer were conducted to quantify airborne biological containing particle concentrations along with dust particles as a function of wind speed and direction over a three week period.

Significant, intermittent enhancements of both non- and bio-fluorescent particles were observed to varying degrees in very specific wind directions and during strong wind events. Analysis of the particle UV induced emission spectra, particle sizes and shapes recorded during these events suggest the majority of particles were likely a subset of dust with weak fluorescence emission responses. A minor fraction, however, were likely primary biological particles that were very strongly fluorescent, with a subset identified as likely being pollen based on comparison with laboratory data obtained using the same instrument.

A strong correlation of biofluorescent particles with wind speed was observed in some, but not all, periods. Interestingly the fraction of fluorescent particles to total particle concentration also increased significantly with wind speed during these events. The enhancement in concentrations of these particles could be interpreted as due to re-suspension from the local ice surface but more likely due to emissions from distal sources within Antarctica as well as intercontinental transport. Likely distal sources identified by back trajectory analyses and dispersion modelling were the coastal ice margin zones in Halley Bay consisting of bird colonies with likely associated high bacterial activity together with contributions from exposed ice margin bacterial colonies but also long range transport from the southern coasts of Argentina and Chile. Dispersion modelling also demonstrated emissions from shipping lanes and as such, marine anthropogenic sources cannot be ruled out. Average total concentrations of total fluorescent aerosols were found to be $1.9 \pm 2.6\,\mathrm{L}^{-1}$ over a 3 week period crossing over from November into December, but peak concentrations during intermittent enhancement events could be up to several 10's $\mathrm{L}^{-1}$. While this short pilot study is not intended to be generally representative of Antarctic aerosol, it demonstrates the usefulness of the UV-

LIF measurement technique for quantification of airborne bioaerosol concentrations, and to understand their dispersion. The potential importance for microbial colonisation of Antarctica is highlighted.

## 1    Introduction

Incursion of biological aerosols (bacteria, fungal spores and pollen) by intermittent inter-continental transport has long been considered an important pathway for potential re-colonisation of the Antarctic biome (Pearce et al., 2016, 2009) in this climate-change sensitive continent. However, the airborne transport and dispersal of biological particles to or within the continent has not been rigorously documented, except from a paleoclimatological perspective. Pearce et al. (2009) suggested that due to the prevailing wind patterns there, it is likely that a proportion of the observed aerobiota will have originated locally. To date, no experiments have attempted to quantify bioaerosol surface emission and deposition fluxes to verify this. Understanding the sources of Antarctic bioaerosols and their dispersion is important for assessing future climate change impacts on the continent's biodiversity. Bioaerosol sources and redistribution mechanisms are also of interest in understanding the contribution to the possible enhancement of climate aerosol-cloud feedback processes in this pristine environment, influencing the evolution of the ice-liquid phase in polar clouds via efficient ice nucleation (Wilson et al., 2015; DeMott et al., 1999) and subsequently impacting radiative feedback responses, e.g. Tan et al. (2016).

### 1.1    Aerobiology of Antarctica

The Antarctic continent is host to a range of active microbial communities which are discussed below. Studies in this region of Antarctica examining the influence of inter-continental transport of biologiocal aerosols were conducted as part of short 2-week studies to catalogue airborne microbial diversity; one in the Austral summer of 2004 and a second in winter of 2005, at the Halley V station. Air masses during these short studies had mostly traversed open sea and land ice near Dronning Maud Land before arriving at the station, but had still spent significant time over Antarctic continental landmasses, especially during easterly winds (Pearce et al., 2009).

Psychrophilic bacteria have been observed in high concentrations in ice samples collected from the Weddell Sea ice edge (Delille, 1992; Helmke and Weyland, 1995). These generally present as rod-like structures approximately 2-3 µm in length; gas vacuole bacteria have also been observed in samples from Antarctic ice-seawater interfaces (IRGENS et al., 1996). It has been suggested that sea-ice melting may alter bacterial availability and hence influence the flux cycle to the atmosphere although this may in turn be reduced by increased bacterial grazing populations (Boras et al., 2010). Individual or aggregates of wind-borne bacteria are generally only transported relatively short distances from their source, however, aeolian dust particles are commonly observed to act as transporters of bacteria, with the potential for their global migration (Yamaguchi et al., 2012; Hallar et al., 2011; Prospero et al., 2005; Griffin et al., 2003). This potential has been highlighted by recent aircraft studies (e.g., Liu et al., 2015).

Diatoms have been observed to be lofted into the atmosphere by bubble bursting and wave breaking processes as the proposed emission mechanism (Cipriano and Blanchard, 1981) and they have been observed in atmospheric samples above sea level

(Harper and McKay, 2010). Diatoms have been observed to act as efficient ice nuclei (Wilson et al., 2015; Knopf et al., 2011; Schnell and Vali, 1976) and elevated ice nucleus concentrations have been reported over subpolar oceanic waters during phytoplankton blooms (Bigg, 1973), suggesting they may play a significant role in modifying cloud microphysical processes at warmer temperatures and in low aerosol concentration environments. Diatom phytoplankton communities are found in cryoconite holes on glaciers (Stanish et al., 2013; Yallop and Anesio, 2010). The holes are formed when wind-blown debris is deposited on the surface of the glacier, causing the surface to melt and form a water filled depression. The debris may contain microorganisms, such as diatoms, and organic material, allowing microbial communities to develop (Stanish et al., 2013). It has recently been demonstrated by Musilova et al. (2016) that biological activity in cryoconite holes may lead to a significant decrease in glacier surface albedo, resulting in enhanced melting and subsequently increasing mass loss.

Penguin guano may also provide a potentially large coastal source of bacteria for airborne redistribution; Zdanowski et al. (2004) identified three major phylogenetic groups (*Pseudomonadaceae*, *Flavobacteriaceae* and *Micrococcaceae*) of bacteria found in bird guano at King George Island, however, little is known about the airborne concentrations and dispersion rates of these microorganisms in this region.

Airborne fungal particles have not been investigated at Halley station to our knowledge. Seasonal airborne spores have been monitored at Signy island, 700 km from the continental Antarctic, in the South Orkney Islands, (60°43'S, 45°36'W) (Marshall, 1996). This work highlighted the possible importance of episodic inter-continental transport of spores as a potentially important contributor to Antarctic biome re-colonisation and ecosystem diversity. The commonest spores found were Ascospores (*Cladosporium conidia*) with daily mean counts ranging from 2.6 to 9.4 x $10^{-6}$ L$^{-1}$. Maximum concentrations were recorded during episodic events likely associated with air-masses from South America. Concentrations could be between 13 to 24 times those of background levels.

Whilst identification of some extremophile microbial populations has been carried out in Antarctica and shown to be dependent on specific air mass trajectory conditions, there has been little in the way of mechanistic studies that quantify concentrations, fluxes or dispersion patterns of these particles once introduced into the continental region (Pearce et al., 2016).

## 2 Methods

### 2.1 Site Description

Aerosol sampling was conducted at the Halley Base Clean Air Sector Laboratory (CASLab) over the period 18 November to 16 December 2015. CASLab was located close to the coast on the Brunt Ice shelf, (Lat. 75°34'59"S, Long. 26°10'0"W), and was approximately 1.1 km SSE of the actual Halley VI research station, approximately 30 m above sea-ice level. It is exposed to the Weddell Sea from the north and west. Winds blow predominantly from the East to West with stronger winds commonly causing re-suspension of dry surface material, with peak winds of $\sim$20 ms$^{-1}$ being observed on several occasions. Average temperature for the sampling period was -6.8°C, with the period from 18-23 November, however, being significantly colder (-11.5°C), than the remaining period average (-5.8°C). The warmest temperature, -1.2°C, was recorded on the 7 December and the coldest, -19°C, recorded on the 19 November.

Pollution from the Halley station diesel generators rarely impacts CASLab due to it being south of the station (off the prevailing wind direction). Furthermore, access is strictly limited to it by foot or by ski. Vehicle access for equipment supply is particularly restricted, and occurs very infrequently (two to three times per year). Any such periods have been excluded from the analysis presented here to minimise contamination artefacts. Due to its isolation, surrounded as it is by "low biomass" snow and ice, most airborne biota are generally considered to have been transported many hundreds of kilometres before reaching the station. However, resuspension of coastal biological containing particles associated with e.g. guano related to bird colonies, Fretwell et al. (2012), as well as from local ice surface sources must also be considered. The nearest ice-free surfaces are the Heimefrontfjella mountain range in East Antarctica, 400 km inland of the Weddell Sea's eastern margin in Western Dronning Maud Land (Jacobs et al., 1996). These extend to over 2000 m above sea level inland and are characterised by very low biomass and biodiversity with no terrestrial vegetation and virtually no birdlife.

The CASLab consists of a stack of three standard 20 foot shipping containers mounted on a steel platform which is raised every 2 years to compensate for snow accumulation and to maintain a constant height above the snow surface. The laboratory is equipped with a stainless steel aerosol inlet comprising a vertical 200 mm i.d. sample stack fitted with a protective snow cowl. Sample air is drawn through the stack by a variable flow fan so as to maintain isokinetic sampling at approximately 240 L min$^{-1}$. Individual instruments are connected to the base of the stack by stainless steel sample lines and these extend well into the main aerosol duct. Further details of the aerosol inlet used in this study are provided in Jones et al. (2008). The effective sampling height for the aerosol measurements in this study was approximately 8 m.

## 2.2 Instrumentation

Fluorescent aerosol number-size distributions were continuously measured using a Wideband Integrated Bioaerosol Spectrometer (WIBS-3D; University of Hertfordshire) on a particle by particle basis. This instrument was designed to identify common bio-fluorophores and discriminate potentially harmful pathogenic bioaerosols from the background population. A full technical description of earlier and later versions of the instrument can be found in Kaye et al. (2005), Foot et al. (2008) and Stanley et al. (2011), while results from monitoring bioaerosols and analysis tools for identification of bioaerosols, mainly at remote sites, can be found in Crawford et al. (2016, 2014), Gosselin et al. (2016), Whitehead et al. (2016), Ziemba et al. (2016), Perring et al. (2015) , O'Connor et al. (2015, 2014), Robinson et al. (2013), Stanley et al. (2011), and Gabey et al. (2013, 2011, 2010). The instrument has an inlet flow of 2.35 L min$^{-1}$, the majority of which is filtered with a HEPA filter to remove all particles, such that the 0.23 L min$^{-1}$ sample flow is sheathed in particle free air to constrain the aerosol into a controlled jet and to minimise contamination of the optics. Aerosol in the sample flow is illuminated by a 635 nm laser and the resultant scattered light is used to determine the particle size and shape using a quadrant detector, where the shape factor (AF) is intepreted as follows: AF < 10-15 is indicative of near spherical particles, AF > 20 aspherical particles, and AF > 30 fibre or rod like particles , where laboratory characterisations using corn starch flour to represent irregular particles and ellipsoidal haematite particles were used as an analogue for rod-like bacterial particles to determine these thresholds (Kaye et al., 2007). The scattering signal is used to sequentially trigger two xenon flash lamps, filtered to output light at 280 and 370 nm, to excite the sample aerosol. Any resultant autofluorescent emissions are collected and filtered into two detection bands (300-400 nm &

420-650 nm) and measured by photomultiplier tubes. This process takes approximately 25 μs and the instrument has a maximum detection rate of 125 particles s$^{-1}$ due to the maximum strobe rate of the flash lamps. This provides three measurements of particle autofluorescence over two excitation wavelengths, particle size and an approximation of particle shape on a single particle basis. The autofluorescence measurements are often referred to as: FL1 (Excitation at 280 nm, Detection at 300-400 nm), FL2 (Excitation at 280 nm, Detection at 420-650 nm), and FL3 (Excitation at 370 nm, Detection at 420-650 nm). The excitation bands of 280 and 370 nm are optimal for excitation of the more common bio-fluorophores, Tryptophan and NADH respectively. The detection wavebands, 300-400 nm and 420-650 nm, also cover the expected bio-fluorophore emission bands of a wide range of other bio-molecular markers (Pöhlker et al., 2012). PBAP of interest (e.g., pollen, bacteria & fungal spores) have been demonstrated to show a detectable autofluorescent response with the WIBS (Savage et al., 2017; Hernandez et al., 2016).

Non-fluorescent particles will exhibit fluorescent signal below the instrument fluorescence threshold, thus the fluorescent signal will be clipped at zero in the processed data as described in Crawford et al., (2015), however, this information and the particle size is still recorded and used to define the non-fluorescent particle population. Non-fluorescent particles are by default classified as non-biological by this technique, however, several non-biological interferents such as mineral dust, brown carbon, diesel soot and cotton fibers have been identified to fluoresce above the commonly used threshold applied here (Savage et al., 2017), thus fluorescent particles may not necessarily be biological in origin and should not be classified as such without further analysis.

Whilst there have been no previous measurements of bioaerosol in the Antarctic using the UV-LIF technique, expected bacteria, such as the common *Pseudomonas spp. (Antarctica)*, have been shown to fluoresce strongly in these wavebands, e.g. the laboratory studies reported by Gabey (2011) as part of the BIO-05 series of experiments where primary biological aerosol particle (PBAP) samples were wet sprayed into the 3.7 m$^3$ NAUA aerosol chamber to be characterised prior to their injection into the 84 m$^3$ AIDA cloud simulation chamber to assess their efficiency as atmospheric ice nuclei (Toprak and Schnaiter, 2013). In addition, laboratory cultures of marine bacteria and algae, that might be expected in this region, also demonstrate tryptophan-like fluorescence (Dalterio et al., 1986; Petersen, 1989), suggesting that the technique is capable of detecting such particles if they are present.

In the WIBS instruments in general a particle is considered to be fluorescent in a given channel (FL1-3), if a threshold fluorescence based on the chamber background mean fluorescence plus 3 standard deviations is exceeded. The WIBS-3D can detect particles with optical diameters between 0.5 to 20 μm, however, due to detector sensitivity and background fluorescence within the optical chamber, the fluorescence of aerosol with diameters $D_p < 0.8$ μm cannot be accurately determined and the counting efficiency decreases at smaller sizes (Gabey et al., 2011). As such all analysis presented here will be limited to aerosols with diameters $D_p \geq 0.8$ μm. We define a particle to be weakly fluorescent if the maximum detector signal in any channel is marginally greater than the applied threshold , e.g., < 20; a moderately fluorescent particle is defined as displaying a maximum fluorescence in any channel in the range of 20-100; similarly, medium fluorescence is defined over a detector range of 100-500 and highly fluorescent as > 500.

The laboratory categorisation and classification of bioaerosols of interest is an ongoing area of research. To date there have been two significant systematic laboratory characterisation studies published using a similar instrument (WIBS-4A); Hernandez et al. (2016) and Savage et al. (2017). We have also performed our own characterisation for the purpose of validating machine learning algorithms experiments (e.g., Ruske et al., 2017; Crawford et al., 2015). The Hernandez et al. (2016) study characterised the autofluorescence of 14 bacterial, 13 pollen and 29 fungal spore samples. The Savage et al. (2017) study characterised 3 bacterial, 5 fungal, 14 pollen, 12 pure biofluorophore, 13 mineral dust, 6 HULIS, 3 PAH, 7 combustion soot and smoke, 3 brown carbon and 3 miscellaneous non-biological particle samples. These studies showed that each particle type demonstrated a broad characteristic autofluorescence, size and asymmetry factor that can be used to interpret and classify ambient measurements, e.g., bacteria were found to predominantly fluoresce in channel FL1 and were generally under 2.5 μm in diameter. While these studies are not exhaustive, the authors note that the fluorescent spectra observed should hold as a broad trend for each particle type. We use such libraries to aide interpretation of our results, along with our own laboratory measurements (provided in Appendix A).

UV-LIF spectrometers such as the WIBS have many advantages over traditional bioaerosol sampling methods, e.g., on-line single particle detection & high time resolution, however, some non-biological fluorescent interferent particles can also show weak auto-fluorescence and so can be a source of false-positives resulting in potential artefacts when interpreting biological materials. This means there can be difficulties discriminating some classes of biological particles unambiguously. Generally the majority of identified interferent non-biological fluorescent aerosols have fluorescence levels similar to the detection limit of the instrument; for example polycyclic aromatic hydrocarbons (PAHs) and PAH containing soot particles of small diameter (< 1 μm) have been demonstrated to fluoresce only weakly in FL1 (Toprak and Schnaiter, 2013; Pöhlker et al., 2012), however we would not expect to see significant concentrations of PAHs or soot particles at this remote site outside of long range transport events. Mineral dusts also contain a small subset of very weakly fluorescent particles due to the presence of luminescence centers within the minerals. These are often associated with rare earth elements, but their observed fluorescent intensity is considerably weaker than observed for primary biofluorophores, Pöhlker et al. (2012). Given the ubiquitous nature of mineral dusts, these weakly fluorescent dust sub-categories may present a significant, even dominant, fraction of recorded fluorescent material, therefore, at the measurement site, particularly during long-range transport events. As such they would likely form their own population clusters, as demonstrated in (Crawford et al., 2016).

At the time of deployment no robust fluorescence calibration method existed for UV-LIF spectrometers. Since this time the first successful calibration methods for WIBS type instruments have become available (Robinson et al., 2017). While the data presented here is uncalibrated, the instrument was routinely sent back to the manufacturer for servicing where no significant changes in the PMT voltages and gains or xenon flash lamp powers were found. At the start of the of measurement period the instrument response was checked with fluorescent doped polystyrene latex spheres to verify the instrument was responding sensibly, however, absolute comparison between calibrations is not possible due to variation in fluorescent intensity between batches of particles and the degradation of their doping material with time. As the unsupervised learning method employed in this study requires no *a priori* information, the lack of calibration should not impact the analysis as the method groups similar data points together for subsequent analysis. The details of this method are described in the next section.

| Cluster | % of $N_{FL}$ | FL1 (a.u.) | FL2 (a.u.) | FL3 (a.u.) | $D_p$ μm | AF (a.u.) | Class |
|---------|------|---------|---------|---------|---------|---------|---------|
| Cl1 | 15.6 | 5.7±20.5 | 4.2±12.8 | 54.9±77.7 | 5.3±3.0 | 24.5±8.1 | Unclassified |
| Cl2 | 2.1 | 135.8±227.4 | 172.1±185.1 | 765.6±535.9 | 7.7±4.0 | 19.9±9.2 | Unclassified |
| Cl3 | 82.1 | 1.9±7.7 | 3.7±8.0 | 6.0±22.5 | 1.3±0.9 | 10.7±4.0 | Dust |
| Cl4 | 0.2 | 678.4±776.8 | 1810.6±222.7 | 1831.3±318.1 | 8.1±5.2 | 18.8±7.7 | Pollen |

**Table 1.** Ward linkage cluster analysis results for the period 18 November to 16 December 2015, showing; the % contribution of the cluster concentration to $N_{FL}$; mean fluorescent intensities in channels FL1, FL2 and FL3; the average optical size, $D_p$ (μm); the average shape factor , (AF - see text); and particle classification, for particles in each cluster.

## 2.3 Data Analysis Methods

In this study we use the approach of Crawford et al. (2016, 2015) for data pre-processing and subsequent cluster analysis. This method has successfully been used to differentiate and identify fungal spores, bacteria and mineral dust classes at remote forests and mountain top sites (Crawford et al., 2016; Gosselin et al., 2016; Whitehead et al., 2016; Crawford et al., 2015). First the data to be clustered were filtered to remove particles with diameters smaller than 0.8 μm and all non-fluorescent particles to leave only fluorescent particles. Due to the paucity of highly fluorescent partilcles we elected to retain saturating particles for clustering to maximise the populations of particles types of interest, e.g., pollens. The fluorescence, size and asymmetry factor single particle data were then normalised using the z-score method prior to clustering, using the Ward linkage. The optimum cluster solution was validated using the Calinski-Harabasz criterion and then integrated time series products were generated for each cluster at 5 minute time resolution. The method used here is described in full in Crawford et al. (2015) and has been compared with other cluster and machine learning techniques by Ruske et al. (2017).

## 3 Results

The single particle data were collected at CASLab during the period 18 November to 6 December 2015. A subset of approximately 17,000 particles were identified as exhibiting fluorescent intensities greater than the fluorescence threshold, which comprised 1.9% of the total number of particles recorded by WIBS, based on particle sizes $D_p \geq 0.8$ μm. The Calinski-Harabasz criterion returned a 4-cluster solution for the Ward linkage. A summary of the resultant cluster centroids is given in Table 1 and their relative contributions to the total aerosol population are presented in Fig. 1.

Cluster 3 (Cl3) was found to be the most dominant based on concentration, representing $\sim 82.1\%$ of the total fluorescent particle population (13,949 particles). Cl3 displays weak fluorescence in all channels. This is consistent with cluster results obtained from previous laboratory and field studies where a subset of mineral dust was identified as the contributor (Crawford et al., 2016; Pöhlker et al., 2012; Gabey, 2011). Particles in this cluster were small, $D_p \sim 1.3$ μm, with an AF value of $\sim 11$, suggesting near spheroidal particles. The AF value reported here is consistent with previous WIBS measurements at an Alpine mountain top site during a Saharan dust event, where a distinct dust cluster was observed with an average AF of $\sim 7$ (Crawford et al., 2016). It is also consistent with several mineral dust samples which were systematically sampled as part of a larger

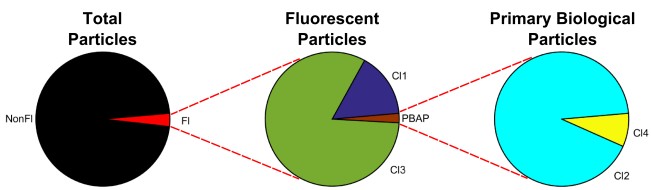

**Figure 1.** The relative proportions of the non-fluorescent, $N_{NonFL}$, and fluorescent, $N_{FL}$, aerosol populations (left); the cluster solution concentrations, Cluster 1 (Cl1, moderately fluorescent), Cl3 (very weakly fluorescent), and PBAP (strongly fluorescent, middle); PBAP here consists of the two clusters, Cl2 and Cl4 (right).

fluorescence characterisation experiment and which displayed average AF values of 10 (Savage et al., 2017). Additionally, this characterisation experiment showed the majority of PBAP samples to display AF values significantly greater than 10.

Cluster 1 (Cl1) is the second most populous cluster, accounting for approximately 15.6% of the total fluorescent particle concentration (2646 particles), but with a much larger average $D_p$ of $\sim$ 5.3 μm, and with AF values of $\sim$ 25. Cl1 particles
are therefore more aspherical. Cl1, interestingly, also shows a moderate fluorescence in FL3, which is significantly different to particles seen in Cluster 3 (Cl3). Without further information it is not possible to identify the actual particles that this fluorescent cluster may represent. We therefore speculate that it is possibly a much larger sized, more fluorescent, sub-population of Cl3, which is segregated from it simply owing to its larger size and asphericity, and is therefore possibly dust. However, this cluster behaviour has not been seen in previous studies and the fluorescence levels are significantly higher than expected. Alternatively
we speculate that this cluster may either be an unidentified large and aspherical primary biological aerosol particle (PBAP), which is UV resistant, or perhaps small UV resistant PBAP attached to a larger dust particle, as described by Yamaguchi et al. (2012). Also, sea salt emitted from open ocean or sea ice (Legrand et al., 2016) could carry PBAP material since both types of region host biological activity, which is known to impact aerosol population (Burrows et al., 2013).

The remaining clusters display significantly greater fluorescence than Cl1 and Cl3. These are more likely representative of
15 larger primary biological aerosols (Hernandez et al., 2016; Crawford et al., 2014; Robinson et al., 2013; Gabey et al., 2010, 2011). Cluster 4 (Cl4, 31 particles), is highly fluorescent in all channels, particularly FL2 & FL3, with mean sizes, $D_p$ of 8.1 μm, and with a mean AF value of 19. They are much larger and less aspherical than generally reported for bacteria containing particles at terrestrial or coastal marine locations, e.g. Harrison et al. (2005). We have conducted a laboratory characterisation study of a small number of bioaerosols previously (at the Defence Science and Technology Laboratory, Porton Down, see
Ruske et al. (2016) for details of the experimental arrangements) with the same instrument used in this study. A cluster analysis of these data revealed that a subset of pollens display very similar fluorescent spectra to those of Cl4 (See Appendix A). This is highly suggestive that Cl4 is representative of pollen, which has been advected to the measurement site via long-range transport, as there is virtually no plant life on the continent. Cl2 (355 particles) is also strongly fluorescent, particularly in FL3, but much less, relatively so, in FL2 compared with Cl4. The Cl2 average $D_p$ was 7.7 μm with an AF of 20, which is very similar to Cl4.
We speculate that they may potentially represent either bacterial aggregates or larger dust particles containing uncharacterised bacteria. The fluorescence spectra do not generally follow those for bacteria or fungal spores observed in previous studies

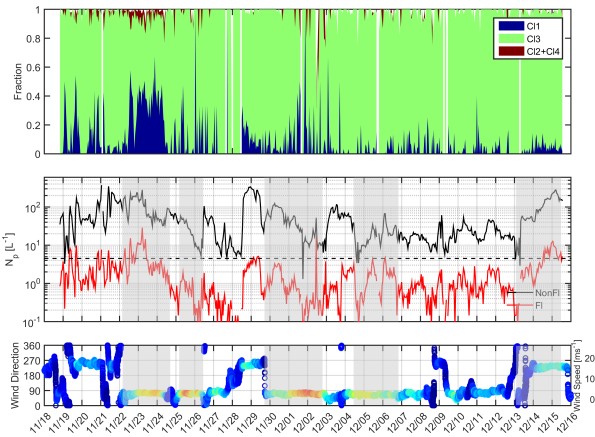

**Figure 2.** Hourly averaged time series of cluster product concentrations (Cl1, Cl2, Cl3 & Cl4 in table 1) to total fluorescent concentration (top); Non-fluorescent, $N_{NonFL}$, and fluorescent particle, $N_{FL}$, concentrations (middle); dashed line indicates overall mean fluorescent value + 1 standard deviation; the corresponding wind direction and speed ($\mathrm{m\,s^{-1}}$) measured at CasLAB (bottom). Grey shaded areas highlight the wind events identified in Table 2.

using the WIBS-3 instrument, which tend to fluoresce most strongly in FL1 (Gosselin et al., 2016; Hernandez et al., 2016; Crawford et al., 2015, 2016). However, in laboratory experiments (using a WIBS-4A) Hernandez et al. (2016) demonstrated that a small subset of certain large fungal spores such as the necrotrophic fungus, *Botrytis*, can fluoresce in all three channels. As such, fungal spores cannot be completely ruled out. Together, these bio-fluorescent clusters contribute approximately 2.3%,

by number, to the total fluorescent particle concentration, so hereafter will be combined into one overall cluster representative of primary biological aerosol particles in our subsequent analyses, and named PBAP cluster (as in Fig. 1).

A time series of the fraction each cluster contributes to the total fluorescent concentration is shown in Fig. 2, along with the corresponding, non-fluorescent, and fluorescent aerosol concentrations and wind data. The average non-fluorescent and fluorescent concentrations over the whole measurement period were $58.8 \pm 66.2\,\mathrm{L^{-1}}$ and $1.9 \pm 2.6\,\mathrm{L^{-1}}$ respectively, with 3.6

$\pm\,2.9\%$ of the total aerosol population being classified as fluorescent.

Periods of significant enhancement, described in detail below, in the fluorescent particle concentration, and clusters Cl1 and PBAP cluster (Cl2+Cl4), were observed to occur during specific high wind events from the north east. These wind events were analysed to identify air mass history.

### 3.1 Wind Driven Fluorescent Enhancement

Significant enhancements in $N_{FL}$, and in particular the ratio of fluorescent particles to total particle concentration, $N_{FL}:N_{TOT}$, occurred mainly during strong NE wind events, which is the most common wind direction at Halley. However, this enhancement was intermittent and did not always occur in these wind sectors, as shown in fig. 3. This may be interpreted in a number of ways. It either suggests depletion of a local surface source due to wind-driven resuspension or more likely due to emission

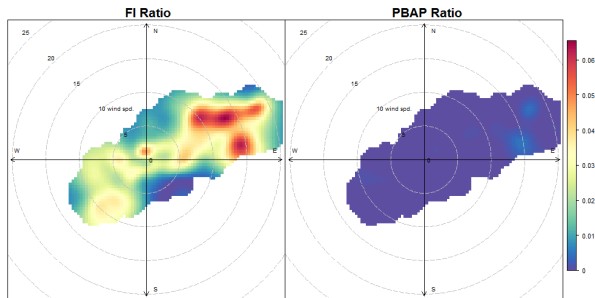

**Figure 3.** Polar plot of the campaign average of the ratio of fluorescent (left panel) and PBAP cluster (right panel) to total aerosol concentration. Polar plots are a function of wind speed and wind direction, with concentric rings representing 5 ms$^{-1}$ increments.

| Wind Event | Period | Wind Speed (ms$^{-1}$) | Wind Direction (°) | $N_{FL}$ (L$^{-1}$) | $N_{Cl1}$ (L$^{-1}$) | $N_{PBAP}$ (L$^{-1}$) |
|---|---|---|---|---|---|---|
| A | 03:00 22/11/2015 - 12:00 24/11/2015 | 11.34±3.73 | 69.19±9.79 | 5.73±7.07 | 2.16±3.53 | 0.34±0.78 |
| B | 03:00 25/11/2015 - 09:00 26/11 2015 | 14.12±2.84 | 72.19±3.01 | 0.67±1.02 | 0.08±0.26 | 0.01±0.10 |
| C | 18:00 29/11/2015 - 22:30 02/12/2015 | 13.43±3.85 | 70.41±4.36 | 1.14±4.61 | 0.06±0.24 | 0.01±0.11 |
| D | 00:00 04/12/2015 - 19:00 06/12/2015 | 11.52±2.07 | 65.04±4.28 | 1.61±1.65 | 0.03±0.16 | 0.01±0.10 |
| E | 12:00 14/12/2015 - 12:00 15/12/2015 | 8.62±1.44 | 230.91±6.06 | 5.83±3.69 | 0.29±0.54 | 0.01±0.13 |

**Table 2.** Summary of highlighted fluorescent particle concentration enhancement and wind event periods, A-E, showing average wind speed, wind direction, concentration of fluorescent particles, $N_{FL}$, concentration of weakly fluorescent particle cluster, Cl1, concentrations of strongly fluorescent particles, PBAP (Cl2+Cl4).

changes in a distal source influencing the sampled air-mass. High fluorescent aerosol concentrations were also observed during SW wind events, however, PBAP cluster concentrations were low during these events.

In the following analysis we have defined an enhanced fluorescence particle concentration event as a period where the total fluorescent particle concentration, $N_{FL}$, is greater than 4.5 L$^{-1}$ (the campaign mean + 1$\sigma$). This threshold was exceeded for approximately 9% of the total measurement period, amounting to 59 hours. We then used periods of enhanced fluorescence, or lack thereof, to define events of interest featuring stable meteorological conditions which we subsequently refer to as wind events, the rationale for each is now briefly described; wind event A is the primary event of interest and features the greatest fluorescent and PBAP cluster concentrations, with high wind speeds from the NE; wind event B features similar meteorological conditions to wind event A, but in contrast to wind event A displays few fluorescent particles; events C & D also feature high wind speeds from the NE and some short fluorescent enhancement events, but low PBAP cluster concentrations; To contrast wind event A, wind event E was chosen to demonstrate flow from the SW and features enhanced fluorescence but low PBAP cluster concentrations. These wind events are summarised in Table 2 along with the mean wind speeds, wind direction, mean fluorescent concentrations, $N_{FL}$, the concentration of the dominant, weakly fluorescent, Cl1 cluster (unclassified) and the concentration of the highly fluorescent (likely biological) PBAP cluster. Peak concentrations of these could however be much higher on shorter timescales within these events which can be more readily detected and quantified by the single particle UV-LIF measurement technique. Whilst strong enhancements in the NE sector were common, these did not always occur,

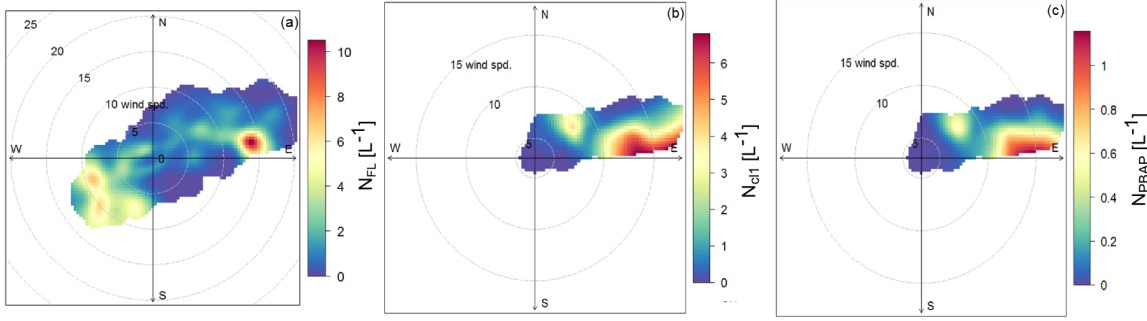

**Figure 4.** Halley CASLab polar concentration plots of total fluorescent particle concentration, $N_{FL}$ (a); weakly fluorescent dust cluster concentration, $N_{cl1}$ (b); and primary biological particle (or biological containing) particle concentration, $N_{PBAP}$ (c), during wind event A. Polar plots are a function of wind speed and wind direction, with concentric rings representing $5\,\mathrm{m\,s^{-1}}$ increments. In each case a strong "hot spot" or possible local "source" might be inferred to the ENE at wind speeds $> 10\,\mathrm{m\,s^{-1}}$, with lesser hot spots seen in the WSW for $N_{FL}$.

hence integrating across all these events for the NE sector can mask the intermittent behaviour seen and the changing relative contributions by the different particle types, e.g., the period 03:00 25/11 - 09:00 26/11 (wind event B) features high wind speeds from the same sector but little to no enhancement is observed suggesting no local sources. Any small changes are likely dominated by distal source variation. Similarly the period 18:00 29/11 - 22:30 02/12 (wind event C) features extended, high

wind speeds from the NE sector, but only 2 short periods of fluorescent particle enhancement were observed. Wind event D (00:00 04/12 - 19:00 06/12) also only shows some minor enhancement. Another period of significant sustained fluorescent particle enhancement is observed between 12:00 14/12 - 12:00 15/12 during a moderately strong wind event, but this time from the west (wind event E). Interestingly the fluorescent characteristics of the particles from this sector were significantly different to wind event A, featuring much lower concentrations of Cl1 and PBAP cluster.

The relationship of $N_{FL}$, $N_{Cl1}$, and $N_{PBAP}$ to wind speed was examined for wind event A, and the results are shown in Fig. 4. The concentrations of Cl1 and PBAP cluster (panels b and c) generally increase with increasing wind speed with a more isolated "hot-spot" for $N_{FL}$ at wind speeds of 12-14 $\mathrm{m\,s^{-1}}$. The highest concentrations of fluorescent aerosol, Cl1 and PBAP clusters occur at wind speeds above $10\,\mathrm{m\,s^{-1}}$ and this persists up to $20\,\mathrm{m\,s^{-1}}$. Weaker enhancements between 5-10 $\mathrm{m\,s^{-1}}$ can be seen in the SW sector in Fig. 4a.

The wind speed dependence for the enhancement during wind event A can be seen more clearly in Fig. 5, where the relationship with wind speed is shown for $N_{FL}$, $N_{Cl1}$, $N_{FL}$:$N_{TOT}$ (the ratio of $N_{FL}$ to the total particle concentration $N_{TOT}$) and $N_{PBAP}$. Interestingly $N_{FL}$, and in particular the ratio of $N_{FL}$:$N_{TOT}$, all start to show enhancement as wind speeds increase above a threshold of 4-6 $\mathrm{m\,s^{-1}}$. This might be interpreted as consistent with surface wind driven re-suspension mechanisms, previously seen in many studies, and therefore suggestive of contributions from more localised ice surface sources for these

particles, as discussed above. This could be the case particularly for the larger particles in Cl1, Cl2 and Cl4. However, this

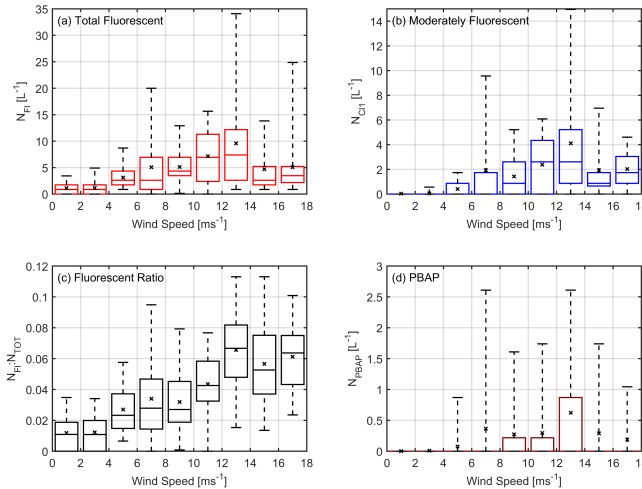

**Figure 5.** Fluorescent particle concentrations as a function of wind speed for the period 22/11/2015 - 25/11/2015 for: (a) Total fluorescent particles, $N_{FL}$; (b) Moderately fluorescent particles, $N_{Cl1}$; (c) Ratio of total fluorescent particles to total particle concentration $N_{FL}:N_{TOT}$ and (d) $N_{PBAP}$ ($N_{Cl2}$+$N_{Cl4}$)

may be fortuitous and the reduction in concentration above $14 \, \mathrm{m \, s^{-1}}$, for $N_{FL}$ and $N_{Cl1}$, should be noted and may be caused by the reduction in inlet transmission efficiency at higher wind speeds. This could suggest a more distal source, supported by the observation that the fluorescence contribution from one of the clusters (Cl4) is likely pollen. This reduction is less obvious for the $N_{FL}:N_{TOT}$ ratio, Fig. 5(c), which is dominated by the much smaller Cl3 particles. The relationship with wind speed

for $N_{PBAP}$ is less clear due to their low concentrations. There is, however, a clear increase in concentration of the larger fluorescent particles in wind event A, for both Cl1 and PBAP clusters. This can be seen in Fig. 6, which compares the campaign averaged particle size distributions for the various clusters for the whole experimental period to the average distributions recorded in each of the wind events, listed in Table 2. Wind event periods A and C show the largest range in the PBAP cluster size distrubutions, whilst events B (Easterly) and E, (Westerly), showed the smallest. However event E did show significant

enhancement in Cl1 and Cl3 concentrations compared to events B, C and D from the Easterly wind sectors. This might suggest a larger source of Cl3 in both sectors but a limited, associated source of PBAP.

### 3.2 Flux Estimates

In the past, short term wind driven enhancements in number concentrations have been used to infer the existence of local surface sources, e.g., Sesartic and Dallafior (2011), however, deriving an aerosol flux from single height concentration measurements

can lead to highly uncertain results (Pryor et al., 2008; Petelski and Piskozub, 2006). If we assume that the majority of the larger fluorescent particles (clusters Cl1 and PBAP cluster) are locally re-suspended then a net flux for these could be estimated using the approach of Sesartic and Dallafior (2011), resulting in fluxes for wind event A of $F_{Cl1} \sim 7.2 \pm 11.8$, and $F_{PBAP} \sim 1.1 \pm 2.6 \, \mathrm{m^{-2} \, s^{-1}}$. However such calculations based on these crude assumptions are very uncertain.

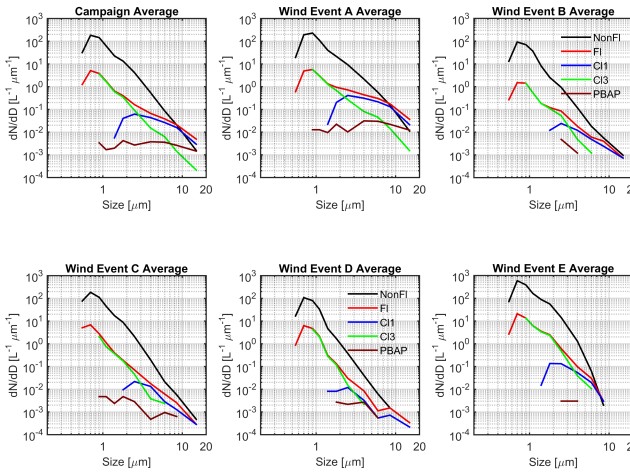

**Figure 6.** Particle size distributions comparing the campaign averaged data (top left panel) with those observed during wind event periods A, B, C, D & E (see Table 2). Black - non-fluorescent particles (NonFl), Red - Total fluorescent particles, Fl; Blue - Cluster 1, (weakly fluorescent particles), Cl1; Green - Cluster 3 very weakly fluorescent particles, Cl3; and Brown - Highly fluorescent particles, PBAP (Cl2+Cl4).

Air mass back trajectory analysis and dispersion modelling (next section) reveal that the aerosol observed during wind event A may be from distal sources, and as such, the local surface source flux estimates presented are invalid in this case. These values are presented to highlight the difficulty in estimating bioaerosol fluxes using these methods, which have previously been used in model assimilations, and they should not be over-interpreted given their uncertainty and may be invalid in this case.

## 3.3 Air Mass Trajectory Analysis

Three-day back trajectory analyses were used for possible source attribution. This used the NOAA HYSPLIT tool, Stein et al. (2015), with 6-hourly averaged re-analysis meteorological data archived at the National Centers for Environmental Prediction-National Center for Atmospheric Research (NCEP- NCAR), with a $2.5°$x $2.5°$ spatial resolution.

Fig. 7 summarises the fraction of time spent over different land classes for the back trajectories to CASLab for each period for the prior 12-120 hours. The land class was specified as being one of three types: Continental (C: land-coastal ice); Open Water (OW: where the sea ice fraction was < 5%); and Sea Ice (SI: where the sea ice fraction was >5%). The fraction of time spent by the air masses below $500$ m altitude is also shown. The periods where the highest concentrations of PBAP cluster occured correspond to those with the largest continental influence within the previous 48 hours. Periods B & E are dominated mainly by sea ice trajectories and show either much lower concentration of PBAP cluster or, in the case of period E, fluorescent particles that exhibited rather different UV-LIF responses.

The UK Met Office Numerical Atmospheric-dispersion Modelling Environment (NAME) model, (Jones et al., 2007) was used to identify particle trajectories during key wind events. This inverse Gaussian plume model approach permits characterisation of emission footprints of air or receptor footprints or tropospheric volumetric flow (in a forward analysis). This

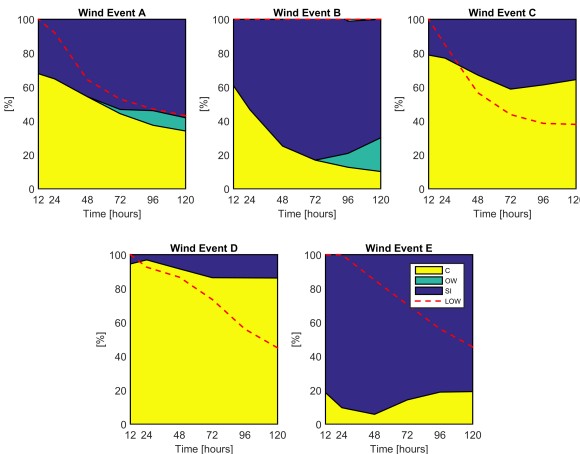

**Figure 7.** Percentage of time spent by an air mass back trajectory, arriving at CASLab, during wind events A-E, as a function of transport time, over different land classes; C: Continental (Yellow: land-coastal ice); OW: Open Water (Green: sea ice fraction < 5%); SI: Sea Ice (Blue: sea ice fraction >5%). LOW shows the proportion of air masses in the history that were < 500 m altitude, (dashed red line).

provides a probablistic interpretation of where the sources of sampled bioaerosol are likely located and how far the particles have travelled. NAME model 5-day back trajectories for periods of interest are shown in Fig. 8.

The top left panel shows particle trajectories that are typical of the period just prior to wind event A, where the majority of particles have passed along the North Dronning Maud Land coastal ice-margin zone and over Neumayer station, prior to arriving at the CASLab via Easterly winds. This behaviour is common for continental circulation patterns here at this time of year. These trajectories pass North and East along the coast via the Lazarev Sea and Lutzow-Holm Bay and eventually from a source also to the South via the Prince Charles Mountains and Mac Robertson Land in East Antarctica.

Wind event A (top right) features the same sources as the prior period, but also displays a second cluster of trajectories from over the northern Weddell Sea, the tip of the Antarctic peninsula, South Shetland Islands and South Orkney Islands, having previously mainly traversed the southern coasts of Argentina and Chile via the Drake passage. Particles consistent with pollen were predominantly observed during this event, suggesting that they have been transported from the coast of South America. This result is consistent with the hypothesis by Pearce et al. (2009), that a significant part of the observed aerobiota may have an external continental source. One conclusion therefore is that the wind driven enhancement of fluorescent aerosols may be due to a combination of resuspension from surface sources, both locally and more distant e.g. likely from along the NE coastal zone, and also from long range transport.

The modelled emissions from wind event B are shown in the middle left panel of Fig. 8 where it can be seen that the majority of particles have originated from within the vicinity of Halley VI station, over the Antarctic peninsula and the Weddell Sea. Notably there are no contributions from eastern continental Antarctica and virtually none from the South American coast. This result is consistent with the HYSPLIT back trajectories, which display a high sea ice land class fraction for wind event B.

Emissions from wind event E (bottom right) show no contibutions from the Weddell sea or Peninsula, but show the majority of particles are local in origin. Coastal eastern Antarctica provides a more distal contribution.

Air mass and particle dispersion analysis has revealed that key periods of interest feature significantly different air mass histories and particle origins. The observation that pollen coincides with particles from the coast of South America reaching the measurement site suggests that long range transport of PBAP may be a significant source of PBAP for the continent, as pollen is otherwise absent during emission events within the Antarctic Circle. Additionally, all wind events except event E display surface emissions from areas of marine traffic to and from the tip of the Antartic Penninsula, thus marine traffic may present a potential minor emission source.

## 4    Summary and Conclusions

We have shown the first results of airborne bio-fluorescent aerosol concentrations recorded by a real-time single particle UV-LIF spectrometer (WIBS) collected in Antarctica. Measurements were collected between 18 November to 16 December 2015 at the Halley Station Clean Air Sector Laboratory (CasLab) near the Halley VI station. Fluorescent particles comprised 1.9% on average of the total aerosol population for particle sizes in the range $0.8 < D_p < 20$ µm, with peak concentrations of up to 65 $L^{-1}$ observed. We adopted a cluster analysis approach to identify and discriminate between different UV-LIF fluorescent aerosol types specific to the instrument used. The resulting cluster concentrations were then analysed with respect to the local meteorological conditions of wind speed and direction and then with respect to air mass histories using HYSPLIT and NAME back trajectory analyses to identify probable sources of these particles.

Fluorescent particle concentration enhancements were observed in NE winds and a strong wind speed dependency for some fluorescent particle clusters was observed. The relationship was less strong for particle clusters that were representative of PBAP due to their much lower concentrations (2.3% of the fluorescent particle population) with one cluster being identified as pollen, and the other as yet unidentified.

A particularly striking feature in the data was the strong wind speed dependence found for the total fluorescent particle fraction. 97.7% of this fraction was dominated by two weakly fluorescent populations, Cl3 and Cl1, in decreasing relative concentrations, with mean sizes for Cl3 of $1.3 \pm 0.9$ µm and for Cl1, $5.3 \pm 3.0$ µm. The range of sizes for these very weakly fluorescent clusters suggests they may be small, naturally fluorescent dust particles, as the fluorescence spectra were consistent with previous studies of long-range transported dust plumes, Crawford et al. (2016). The Cl1 cluster showed the largest asphericity factor which also supports this.

The highly fluorescent particles represented by Cl2 and Cl4 are likely biological, based on comparison with laboratory studies. Specific identification remains tentative, however in case of Cl4 (the smallest contributor to fluorescent particle number concentration), we can suggest this was a pollen class (see appendix A). Cluster 2, however, remains unknown and has not been observed previously, either in laboratory studies or in ambient air experiments. We speculate that this population may represent moderately fluorescent primary biological particles (e.g. UV resistant or particles with low metabolic activity), bacterial aggregates or possibly biological particles such as bacteria associated with larger dust particles during long range transport,

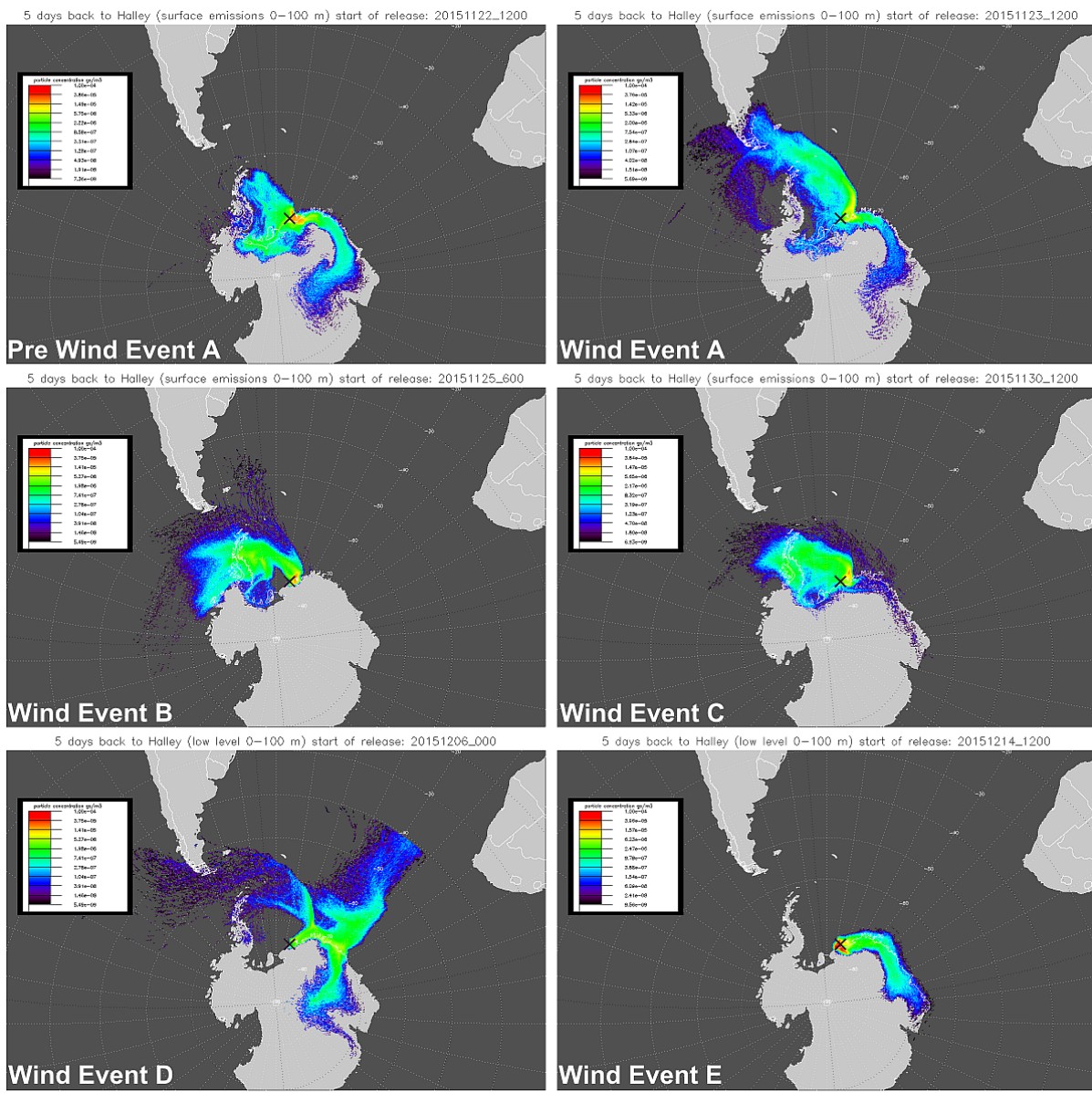

**Figure 8.** 5-day back trajectory analysis using the NAME particle dispersion model, with surface emissions for; one day prior to wind event A (top left); wind event A (top right); wind event B (middle left); wind event C (middle right); wind event D (bottom left); and wind event E (bottom right), for altitudes < 100 m. X marks the location of the Halley VI station.

given the relatively large size and asphericity factor of this cluster. While there are numerous sources of bacteria in the region (see section 1.1), no bacterial cluster signatures were observed, based on the laboratory samples currently available. This suggests that airborne concentraction of these bacteria are either well below the detection limit of the instrument or that they have significantly different autofluorescence signatures to the laboratory samples.

These different observations are likely the net result of the different air-mass sources identified. Whilst local resuspension fluxes can be estimated and were found to be consistent with modelling estimates based on filter sample collections in the Arctic (Sesartic and Dallafior, 2011), these are highly uncertain due to the methodology adopted in such studies for such environments.

    The wind speed enhancements might suggest that a significant source of these fluorescent particles possibly exists on or in
the local ice surface in the region ENE of the CASLab site, but are more likely to have been transported from distal sources, e.g., the South American continent, and the dispersion model supports this as the more likely scenario. The presence of particles characteristic of pollen is evidence towards the latter conclusion, however, the dispersion model results also display emissions from regions which feature significant marine traffic, and thus anthropogenic marine sources for these observations cannot be ruled out. Only a more detailed and robust micrometeorological flux closure approach coupled with multiple site measurements
within the key source footprint regions can confirm this.

    The real-time, single particle UV-LIF technique used in this case study has been demonstrated as a useful method for detecting aerobiota in the low concentration Antarctic environment. The continual improvement in detection capacity and sensitivity of UV-LIF instruments could eventually provide useful information as part of a long term monitoring strategy for understanding the biodiversity changes in these remote ice dependent refugia. We suggest that further long term studies with supporting
offline measurements are needed to build up a climatology of bioaerosol events to better understand bioaerosol concentrations and long range transport in the general case.

## Appendix A: Laboratory Characterisation of Fluorescent Particles

| Cluster | % of $N_{FL}$ | FL1 (a.u.) | FL2 (a.u.) | FL3 (a.u.) | $D_p$ μm | AF (a.u.) |
|---|---|---|---|---|---|---|
| Cl1 | 31.7 | 10.8±65 | 157.1±212 | 315.2±341.2 | 3.4±2.3 | 21.2±9.6 |
| Cl2 | 68.3 | 322.3±417.1 | 1741.8±350.8 | 1830.4±273.6 | 11.8±3.2 | 15.4±7.4 |
| Cl4 (ambient) | - | 678.4±776.8 | 1810.6±222.7 | 1831.3±318.1 | 8.1±5.2 | 18.8±7.7 |

**Table A1.** Ward linkage cluster analysis results for pollen laboratory samples, showing; the % contribution of the cluster concentration to $N_{FL}$; mean fluorescent intensities in channels FL1, FL2 and FL3; the average optical size, $D_p$ (μm); and the average shape factor (arb. units), for particles in each cluster. Ambient cluster Cl4 from table 1 shown for comparison.

    A small selection of bioaerosols and fluorescent material were sampled with the WIBS-3D in a series of laboratory char-
acterisations at the Defence Science and Technology Laboratory Porton Down facility prior to its deployment in Antarctica.

Particles of interest were separately aerosolised into a large, clean HEPA filtered containment chamber (incorporating a re-circulation fan), from which the WIBS-3D drew measurement samples. Dry materials were aerosolised directly from small quantities of powder using a filtered compressed air jet. The sample chamber was cleaned with absorbent paper and sodium hypochlorite in between samples to minimise contamination (Ruske et al., 2017). Four typical pollens (birch, paper mulberry, ragweed and rye grass) were selected from the sample set and clustered using the HCA method described in section 2.3. The pollens selected are common allergens in the UK are readily available from commercial suppliers. This yielded a two cluster solution, as described in table A1. The major cluster, Cl2, accounts for ~70% of the fluorescent material, suggesting that this cluster is generally representative of the sampled pollens. This cluster features mean fluorescent intensities, size and shape factors which are very similar to that of ambient cluster 4 observed at Halley (see table 1), with high fluorescent intensities observed in FL2 and FL3 and mean particle sizes of approximately 10 μm. This is highly suggestive that ambient cluster Cl4 is representative of pollen. Cluster Cl1 is most likely the result of sampling pollen which has been fragmented during aersolisation (e.g., Savage et al., 2017).

*Author contributions.* I. Crawford wrote the paper and performed analysis; M.W. Gallagher and T.W. Choularton were project managers and contributed towards data interpretation; K.N. Bower and M.J. Flynn conducted the field experiment; S. Ruske contributed towards data analysis; C. Listowski and N. Brough provided CASLab meteorological data and field support; Z. Flemming provided the NAME analysis; V.E. Foot provided supporting laboratory data; W.R. Stanley provided WIBS data support.

*Competing interests.* The authors declare that they have no conflict of interest.

*Acknowledgements.* We would like to thank the University of Manchester and British Antarctic Survey teams conducting the "Microphysics of Antarctic Clouds" project , Dr. Tom Lachlan-Cope and Prof. Tom Choularton, supported by NERC grant NE/K0142X/1, for kindly including the WIBS measurements as part of their experiment. We would also like to thank the Dstl team for loan of the WIBS instrument and for access to their laboratory databases to support the UV-LIF aerosol interpretation. We thank Prof. Paul Kaye for his continued support of the WIBS. We would finally like to thank the BAS team at Halley VI for all their support. The authors gratefully acknowledge use of the NOAA Air Resources Laboratory (ARL) for the provision of the HYSPLIT transport and dispersion model used in this publication. S. Ruske is in receipt of a NERC DTP studentship.

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
