# Peer review of "Real Time Detection of Airborne Fluorescent Bioparticles in Antarctica"

_Atmospheric Chemistry and Physics, 2017_

## Referee Comment (RC1) · Anonymous Referee #1 · 15 Jun 2017

Important contribution, worthy of publication

1. Calibration and classification of bio-particles is required (minimum theoretically, but possible practically) 'Bio-particles' misleading as measuring fluorescence, most of which may be down to biological origin, but by no means all Title: Real time detection of fluorescent particles in Antarctica P4 L22 Needs calibrating for different bio-particles as 2.35 L min-1 is a low flow rate P4 L30 What about non fluorescent bio-particles? P5 L16-17 This sentence strongly suggests that UV-LIF needs proper calibration for bio-particles

2. P4 L1 'near-sterile' is not appropriate as it cannot be substantiated, use 'low biomass'

3. Further methodological detail required P4 L16 'The instrument was designed to identify common fluorophores' detail needed here as fundamental to what is being measured P4 L22 Filtered – how, what proportion of bio-particules is removed by filtration? P5 L3 Many more bacteria are common aerosols, a diverse range of examples could be tested P5 L1 This needs more detail in order for the reader to be able to repeat the approach P16 L1 What was the rationale for these pollen types?

4. Further contextual detail helpful P5 L16 Specify what these 'many advantages' are?

5. Minor issues and typos P5 L3 Genus and species names in italics P5 L3 Capital A for Antarctica

---

## Referee Comment (RC2) · M. T. Könemann (Referee) · 15 Jun 2017

Synopsis (Crawford et al.)

-Accept, minor revision-

The manuscript by Crawford et al. entitled "Real Time Detection of Airborne Bioparticles in Antarctica" presents the results of short-term measurements with a Wideband Integrated Bioaerosol Sensor (WIBS, Model 3D) at the Halley Base Clean Air Sector Laboratory (CASLab) during Antarctic Summer in 2015. Data were collected within a three-week period and subsequently analyzed using a proven pre-processing- and data clustering approach specified in Crawford et al., 2015, 2016. Additionally, geospatial- and meteorological analyses were performed for back- and source-tracking of potential

primary biological aerosol particles (PBAPs) and non-biological particles like dust. The authors state the following major findings:

I. On average, fluorescent particles comprise 1.9 % out of the total aerosol concentration (in a size range between 0.8 and 20 $\mu$m).

II. Two clusters were classified as dust particles (Cl3) and pollen (Cl4). Cluster Cl1 and Cl2 remain unclassified.

III. For some events, the fluorescent particle concentration seems to be strongly correlated to wind speed and/or wind direction.

IV. Pollen may undergo long-range transport from the coast of Southern America.

Even if commercially available instruments for laser/light-induced fluorescence detection (e.g. WIBS, UV-APS) are commonly used in the bioaerosol community for over 10 years, assessment of physical and technical instrument properties, data analyses and interpretation are still quite challenging. The current manuscript is well written and represents a useful data set out of a unique environment and, therefore, contributes an additional "piece in the puzzle" for a better understanding of aerosol dynamics and data analyses in the future. However, I have some comments/suggestions regarding data acquisition and interpretation which I will explain in detail in the following sections.

Specific Comments:

I. Short-term measurements with a single instrument in a complex environment with rather unknown atmospheric Dynamics

As stated above, the use of LIF instruments is highly challenging and we're currently not even able to clearly explain (bio)aerosol dynamics in environmental systems right on our own doorstep. Especially therefore, measurements over a duration of roughly one month in Antarctica, with it's very low particle concentrations, will most likely lack statistical relevance to some extent. Additionally, only a single instrument was used for data acquisition without a point of reference in the form of other on- (e.g. an Optical

Particle Sizer, OPS) or off-line (e.g. impactor) techniques to countercheck derived data from the WIBS-3D to i.) verify data accuracy and ii.) support results out of the cluster classification approach. Even if the authors refer to measurements with the same device prior to the campaign in Antarctica (page 15, line 24), the reader has to "trust" the measurement accuracy of the WIBS-3D used in this study. A simple, e.g., glass slide impactor for some quick microscopic analyses would have had improved the overall quality, especially by supporting cluster classifications.

II. Wind speed and inlet kinetics

Wind speeds on site ranging from 8.62 to 14.12 ms-1 (table 2, page 8). At such high rates, inlet kinetics becomes serious business. However, the flow rate of the bypass used (flow fan) is not stated, which becomes a critical factor for concentration- and size cutoffs. In general, the whole inlet system may need to be described a bit more in detail (e.g. was a diffusion or Nafion dryer used in between?). To me, figure 4, page 10 serves as an indicator for a potential sampling cutoff, where particle concentrations are decreasing above $\sim$ 14 ms-1. Therefore, it seems to me that the flow rate of the bypass was too low to force particles onto a bow-trajectory at such high wind speeds. Long story short: I think that particles at such wind speeds just flew over the inlet horizontally, not reaching the WIBS.

III. Wind speed and snow/ice Crystals

Temperatures mostly below zero and high wind speed rates lead me to the thought in how far ice crystals from local sources may contribute to the measured data set. To me, it seems to be reasonable that, at least, a minor portion of particle concentrations counted, may be ice crystals. Furthermore, crystal structures on particle surfaces may also affect the asymmetry factor (and also sizing) by changing light scattering patterns detected by the Quad-PMT. However, the occurrence of ice crystals depends on the overall inlet system which needs, as stated above, a more detailed description.

IV. Vessels as potential emission sources

Even if the marine traffic in this particular area is considered to be rather low, vessels as a potential particle emission source has to be kept in mind though. Attached is a link showing a traffic density map from 2015 (Click on density map button on left):

https://www.marinetraffic.com/en/ais/home/centerx:-59.2/centery:-64.6/zoom:4

As you can see in here, there is a main traffic route in NW direction including mostly tankers, cargo- and fishing vessels. Compared to the back-trajectory analyses in figure 7 (page 14), all wind events (except for E) crossed or brushed the main traffic route for which I think that it has to be considered as a potential emission source to some extent.

V. Geospatial analyses

The data processing of figure 6 (page 12) is unclear to me and needs some further explanation. How were the land class types in combination with back-trajectories processed? Was the trajectory length used? Or was the trajectory "footprint" put onto a, e.g., raster map and blanked out?

Technical corrections: Single trajectory plots in figure 7 (page 14) need captions for better allocation.

Final comment: The current manuscript provides an interesting data set and will be useful for the whole bioaerosol community and should, therefore, be published. However, the authors need to state the general "case study-nature" of the manuscript more clearly and discuss effects and potential interferences which might occur in this complex environment (e.g. snow and ice, vessels) more detailed. Furthermore, the inlet system used in this study needs some further description.

---

## Referee Comment (RC3) · Anonymous Referee #3 · 22 Jun 2017

This manuscript presents measurements of fluorescent aerosol made in Antarctica over a period of weeks using a WIBS. Although the authors find that fluorescent particles are a minor component (a few %) of total aerosol, there are some interesting features in the data worthy of publication. WIBS data is analyzed using a clustering method previously published by this group and 4 component populations are identified. Two of these clusters (together accounting for >97% of the fluorescent aerosol) are only weakly fluorescent and are hypothesized to be non-biological fluorescent aerosol, possibly dust. The other two clusters have more fluorescent intensity, are hypothesized to be biological and one of these is very similar to a cluster identified from laboratory samples of various pollen. Fluorescent loadings are analyzed as a function of wind speed for specific periods of interest and the authors state that high levels of fluores-

cent aerosols were primarily (though not always) associated with flow from the NE. Back trajectories are also analyzed and the authors posit that fluorescent aerosol (and thus pollen) arrives at the site as a result of long range transport from as far away as South America.

Comments:

This paper presents the first fluorescent aerosol observations reported for Antarctica and, as such, it is a worthy contribution to the literature. However I found portions confusing and also recommend including more information in certain places. Much of my discomfort arises from the fact that the 5 periods of interest seem rather arbitrary, at least given the information presented. As such, I don't know how to interpret observed differences between these periods or what they mean for fluorescent aerosol in Antarctica more generally. Specific suggestions for improvement are included below.

1. With any discussion of intensity of fluorescent signals the question of calibration arises. I appreciate that there is, as yet, no widely-accepted calibration for fluorescence in the WIBS and it seems that this instrument has been used in numerous laboratory and field studies without significant intentional modification. There is some discussion of this in the discussion of pollen identification (i.e. that the same instrument was used to look at pollen samples and they look very similar in intensity to Cl4) however it would be appropriate to include a more thorough related discussion in the methods section. Do the authors have any information regarding the stability of fluorescent intensity measurements over time? Are the instrument gains used here the same as those used in previously published work from this group? Can you comment on whether or how changes or instability in fluorescent sensitivity would affect the clustering algorithm? Can the authors comment on what kinds of laboratory-generated particles they have observed to fall into the weakly, moderately, medium and highly fluorescent populations? Those categories seem arbitrary and are used only minimally in the subsequent analysis.

2. On a related note, it would be good to include the numbers of particles sampled that fell in each cluster and also the number of particles that saturated the detector. Do the detectors for this WIBS saturate around 2000 counts? If so, given that the stated average intensity in the pollen population is ∼1800+/-300 after exclusion of saturating particles, it seems that a substantial fraction of pollen particles would saturate and you might be underestimating the contribution of that population.

3. A relatively minor point but, in your discussion of asymmetry factor, I believe dust is typically quite fractal (e.g. Bi, Huang et al, ACP 2016 or Yu, Zhu, et al, ACP 2015) yet your dust cluster AF indicates relatively sphere-like. Can the authors provide information to bolster confidence in the retrieved AF from the WIBS? (i.e. any data from calibrations with known aspherical particles or any corroborating reports of relatively spherical dust?)

4. I am confused by the discussion surrounding the wind events. First, the authors define a level above which they consider fluorescent concentrations elevated and imply that they are going to look at periods where that happened. Then, however, two of the five periods in table 2 don't have elevated fluorescent concentrations (the 2nd and the 4th) while there are periods that seem to have elevated fluorescent concentrations that are not included in the analysis (i.e. early on in the project and on 11/29). Is the selection driven mainly by wind speed and direction? Why include the 5th period and not periods from 20-21 and 29 Nov? Are these just meant to be case studies of the different combinations of wind and aerosol loadings observed? Please clarify how these 5 periods of interest were chosen. It would also be helpful if these periods were marked in Figure 2 so that the reader doesn't have to mentally combine the table and the figure.

5. In the text, the authors state that high levels of fluorescent aerosols were primarily associated with flow from the NE but I don't think this statement is supported by the data presented. To me it seems that there was one period of fluorescent enhancement from the NE and one from the W. There are possibly even two instances of high

loadings with westerly flow if you consider the noisy but relatively elevated concentrations at the beginning of the project in addition to what was seen in the 5th highlighted period. Other instances of flow from either direction don't necessarily bring elevated concentrations and I don't know what the explanation is for this behavior but I don't think it's as clean as currently presented.

6. The authors also state that they see enhancements in the ratio of fluorescent to total aerosol at particular times. It is nearly impossible to assess this ratio from the graph presented. I recommend adding a panel or a figure to show a time series of the fluorescent fraction, possibly showing two traces where one shows the "dust-like" fluorescent fraction and one shows the PBAP fluorescent fraction. 7. I don't fully understand Figure 3. Was this made from the average of all periods when the wind was from the NE and, if so, how was this average calculated? How is it that the plot for total fluorescent particles has a component in the SW quadrant but the other two do not? In panel b, it is labeled as dust but also as Cl1. I thought dust was Cl3 and Cl1 was unclassified. Either way, why show the plot for one but not the other? 8. In the caption of Fig 4 it is stated that these plots are only for the NE wind event with the highest fluorescent loadings however the text on lines 1-3 of the same page implies that it is for all of the selected events. Please make these consistent. If the graphs are really only for a single event, it would be interesting to know whether similar behavior was observed during other periods. What does it look like if similar graphs are made for the westerly event that had relatively high fluorescent loadings?

9. I am not well-versed in calculations of flux, and I cannot speak to the validity of the method used here. In any case, I don't really see the point of calculating a flux under the present circumstances. If the elevated concentrations are episodic and not systematically associated with a particular flow direction or meteorological context, then it doesn't seem that this is likely to represent flux from sea ice or the ocean or any other dispersed source but, rather, will represent flux from a particular but unidentified bioaerosol source at an unknown location and I don't see the utility. Flux from the local

environment might be better assessed by looking at wind events without elevated fluorescent concentrations but, again, I don't know enough about flux to know if this would be robust or even possible.

10. With regard to the airmass trajectory analysis, it would be nice to see maps for all of the events discussed. Was event A the only time that flow arrived from S. America or was there a time with similar back trajectories but little fluorescent aerosol enhancement?

11. As stated above, much of my discomfort with this paper arises from the fact that so much of the discussion centers on analysis of 5 events (and of those 5, only one or two get much attention) and the selection of these events is unclear to me. It is therefore difficult to develop a sense for how representative they might be, how to interpret the variability between them or what they mean in a larger context. The text is often written as though systematic relationships have been found which I find a bit misleading given that the study duration was relatively short and these "relationships" are extrapolated from single events. I recommend rephrasing these statements and revisiting the data analysis to more clearly delineate the observations themselves, the generalizations made based on the observations and the limitations to these generalizations imposed by the short duration of the study and the episodic nature of the environment.

---

## Author Comment (AC1) · 9 Aug 2017

For clarity, the referee's comments are copied in black and our responses are offset in blue.

We thank the reviewer for their helpful comments and recommendations which we address below.

Important contribution, worthy of publication.

1. Calibration and classification of bio-particles is required (minimum theoretically, but possible practically) 'Bio-particles' misleading as measuring fluorescence, most of which may be down to biological origin, but by no means all Title: Real time detection of fluorescent particles in Antarctica

We do not feel that the title is misleading.  We clearly state our conservative estimates of bioparticle concentrations and how they were classified.

P4 L22 Needs calibrating for different bio-particles as 2.35 L min-1 is a low flow rate

The sheath flow is filtered using a HEPA filter which will remove all particles at this flow rate.

P4 L30 What about non fluorescent bio-particles?

PBAP of interest (e.g., pollen, bacteria & fungal spores) have been demonstrated to show a detectable autofluorescent response with the WIBS (Hernandez et al., 2016, Savage et al., 2017).  Non-fluorescent particles will exhibit fluorescent signal below the instrument fluorescence threshold, thus the fluorescent signal will be clipped at zero in the processed data as described in Crawford et al., (2015), however, this information and the particle size is still recorded and used to define the non-fluorescent particle population.

P5 L16-17 This sentence strongly suggests that UV-LIF needs proper calibration for bio-particles

We assume that the referee is referring to the requirement for a training library of autofluorescent signatures for comparative attribution, rather than a calibration for fluorescent intensity in our answer.

The laboratory categorisation of bioaerosols of interest is an ongoing area of research.  To date there have been two significant systematic laboratory characterisation studies published using a similar instrument (WIBS-4A); Hernandez et al., (2016) and Savage et al., (2017).  We have also performed our own characterisation for the purpose of validating machine learning algorithms experiments (e.g., Ruske et al., 2017 & Crawford et al., 2015).

The Hernandez et al., (2016) study characterised the autofluorescence of 14 bacterial, 13 pollen and 29 fungal spore samples.  The Savage et al., (2017) study characterised 3 bacterial, 5 fungal, 14 pollen, 12 pure biofluorophore, 13 mineral dust, 6 HULIS, 3 PAH, 7 combustion soot and smoke, 3 brown carbon and 3 miscellaneous non-biological particle samples.  These studies showed that each particle type demonstrated a broad characteristic autofluorescence, size and asymmetry factor that can be used to interpret and classify ambient measurements.  We use such libraries to aide interpretation of our results, along with our own laboratory measurements, such as those provided in the supplementary material.

2. P4 L1 'near-sterile' is not appropriate as it cannot be substantiated, use 'low biomass'

We will revise this as suggested.

3. Further methodological detail required. P4 L16 'The instrument was designed to identify common fluorophores' detail needed here as fundamental to what is being measured

This is elucidated on P4 L34, briefly; FL1 is optimal for the detection of tryptophan and proteins; FL3 is optimal for NADH detection as described in Kaye et al. (2005).

P4 L22 Filtered – how, what proportion of bio-particles is removed by filtration?

This refers to the filtration of the sheath flow. This is filtered with a HEPA filter to remove all particles, such that the 0.23 L/min sample flow is sheathed in particle free air to constrain the aerosol into a controlled jet and to minimise contamination of the optics. As such, none of the sample flow aerosol is removed.

P5 L3 Many more bacteria are common aerosols, a diverse range of examples could be tested

The laboratory categorisation of bioaerosols of interest is an ongoing area of research. To date there have been two significant systematic laboratory characterisation studies published using a similar instrument (WIBS-4A); Hernandez et al., (2016) and Savage et al., (2017, under review). These studies cumulatively sampled 16 different bacterial samples and found that each predominantly fluoresces in channel FL1 and were generally under 2.5 μm in diameter. While these studies are not exhaustive, the authors note that the fluorescent spectra observed should hold as a broad trend for each particle type.

P5 L1 This needs more detail in order for the reader to be able to repeat the approach

The details are provided in Gabey (2011) and Toprak & Schnaiter (2013). We will include a reference to the latter and update the text to the following:

"Whilst there have been no previous measurements of bioaerosol in the Antarctic using the UV-LIF technique, expected bacteria, such as the common Pseudomonas spp. (Antarctica), have been shown to fluoresce strongly in these wavebands, e.g. the laboratory studies reported by Gabey (2011) as part of the BIO-05 series of experiments where PBAP samples were wet sprayed into the 3.7 m$^3$ NAUA aerosol chamber to be characterised prior to their injection into the 84 m$^3$ AIDA cloud simulation chamber to assess their efficiency as atmospheric ice nuclei (Toprak & Schnaiter, 2013)."

P16 L1 What was the rationale for these pollen types?

The pollens selected are common allergens in the UK are readily available from commercial suppliers.

4. Further contextual detail helpful P5 L16 Specify what these 'many advantages' are?

We will update the sentence to the following to include the requested details:

UV-LIF spectrometers such as the WIBS have many advantages over traditional bioaerosol sampling methods (e.g., on-line single particle detection & high time resolution)…

5. Minor issues and typos P5 L3 Genus and species names in italics P5 L3 Capital A for Antarctica

We will correct this in the revised manuscript.

**References**

Crawford, I., Ruske, S., Topping, D. O., and Gallagher, M. W.: Evaluation of hierarchical agglomerative cluster analysis methods for discrimination of primary biological aerosol, Atmos. Meas. Tech., 8, 4979-4991, https://doi.org/10.5194/amt-8-4979-2015, 2015.

Gabey, A. M.: Laboratory and field characterisation of fluorescent and primary biological aerosol particles, Ph.D. thesis, University of Manchester, 2011.

Hernandez, M., Perring, A. E., McCabe, K., Kok, G., Granger, G., and Baumgardner, D.: Chamber catalogues of optical and fluorescent signatures distinguish bioaerosol classes, Atmos. Meas. Tech., 9, 3283-3292, https://doi.org/10.5194/amt-9-3283-2016, 2016.

Ruske, S., Topping, D. O., Foot, V. E., Kaye, P. H., Stanley, W. R., Crawford, I., Morse, A. P., and Gallagher, M. W.: Evaluation of machine learning algorithms for classification of primary biological aerosol using a new UV-LIF spectrometer, Atmos. Meas. Tech., 10, 695-708, https://doi.org/10.5194/amt-10-695-2017, 2017.

Savage, N., Krentz, C., Könemann, T., Han, T. T., Mainelis, G., Pöhlker, C., and Huffman, J. A.: Systematic Characterization and Fluorescence Threshold Strategies for the Wideband Integrated Bioaerosol Sensor (WIBS) Using Size-Resolved Biological and Interfering Particles, Atmos. Meas. Tech. Discuss., https://doi.org/10.5194/amt-2017-170, in review, 2017.

Toprak, E. and Schnaiter, M.: Fluorescent biological aerosol particles measured with the Waveband Integrated Bioaerosol Sensor WIBS-4: laboratory tests combined with a one year field study, Atmos. Chem. Phys., 13, 225-243, https://doi.org/10.5194/acp-13-225-2013, 2013.

---

## Author Comment (AC2) · 9 Aug 2017

**M. T. Könemann (Referee #2)**

For clarity, the referee's comments are copied in black and our responses are offset in blue.

Synopsis (Crawford et al.)

-Accept, minor revision-

The manuscript by Crawford et al. entitled "Real Time Detection of Airborne Bioparticles in Antarctica" presents the results of short-term measurements with a Wideband Integrated Bioaerosol Sensor (WIBS, Model 3D) at the Halley Base Clean Air Sector Laboratory (CASLab) during Antarctic Summer in 2015. Data were collected within a three-week period and subsequently analysed using a proven pre-processing- and data clustering approach specified in Crawford et al., 2015, 2016. Additionally, geospatial and meteorological analyses were performed for back- and source-tracking of potential primary biological aerosol particles (PBAPs) and non-biological particles like dust. The authors state the following major findings:

I. On average, fluorescent particles comprise 1.9 % out of the total aerosol concentration (in a size range between 0.8 and 20 μm).

II. Two clusters were classified as dust particles (Cl3) and pollen (Cl4). Cluster Cl1 and Cl2 remain unclassified.

III. For some events, the fluorescent particle concentration seems to be strongly correlated to wind speed and/or wind direction.

IV. Pollen may undergo long-range transport from the coast of Southern America.

Even if commercially available instruments for laser/light-induced fluorescence detection (e.g. WIBS, UV-APS) are commonly used in the bioaerosol community for over 10 years, assessment of physical and technical instrument properties, data analyses and interpretation are still quite challenging. The current manuscript is well written and represents a useful data set out of a unique environment and, therefore, contributes an additional "piece in the puzzle" for a better understanding of aerosol dynamics and data analyses in the future. However, I have some comments/suggestions regarding data acquisition and interpretation which I will explain in detail in the following sections.

We thank the reviewer for their helpful comments and recommendations which we address below.

**Specific Comments:**

I. Short-term measurements with a single instrument in a complex environment with rather unknown atmospheric Dynamics As stated above, the use of LIF instruments is highly challenging and we're currently not even able to clearly explain (bio)aerosol dynamics in environmental systems right on our own doorstep. Especially therefore, measurements over a duration of roughly one month in Antarctica, with it's very low particle concentrations, will most likely lack statistical relevance to some extent. Additionally, only a single instrument was used for data acquisition without a point of reference in the form of other on- (e.g. an Optical Particle Sizer, OPS) or off-line (e.g. impactor) techniques to countercheck derived data from the WIBS-3D to i.) verify data accuracy and ii.) support results out of the cluster classification approach. Even if the authors refer to measurements with the same device prior to the campaign in Antarctica (page 15, line 24), the reader has to "trust" the measurement accuracy of the WIBS-3D used in this study. A simple, e.g.,

glass slide impactor for some quick microscopic analyses would have had improved the overall quality, especially by supporting cluster classifications.

> This was an opportunistic pilot study in the region to assess the utility of the technique as part of a larger airborne experimental campaign which had very different scientific goals and objectives (Microphysics of Antarctic Clouds, MAC).  As such, while other online aerosol instrumentation was running at the site, they were configured to detect nucleation burst events at much smaller sizes to support the cloud microphysics measurements.  We agree that glass slide/impactor samples would have been of great benefit to the analysis and filter samples were taken during airborne operations from 01/12/15 onwards, however, no such samples were taken during wind event A where the majority of PBAP/pollen was observed.

**II. Wind speed and inlet kinetics**

Wind speeds on site ranging from 8.62 to 14.12 ms-1 (table 2, page 8). At such high rates, inlet kinetics becomes serious business. However, the flow rate of the bypass used (flow fan) is not stated, which becomes a critical factor for concentration- and size cutoffs. In general, the whole inlet system may need to be described a bit more in detail (e.g. was a diffusion or Nafion dryer used in between?). To me, figure 4, page 10 serves as an indicator for a potential sampling cutoff, where particle concentrations are decreasing above $\sim$ 14 ms-1. Therefore, it seems to me that the flow rate of the bypass was too low to force particles onto a bow-trajectory at such high wind speeds. Long story short: I think that particles at such wind speeds just flew over the inlet horizontally, not reaching the WIBS.

> We thank the referee for their useful comments.  We will include more detail on the sampling arrangement used and we will include a short discussion on the potential for reduced sampling efficiencies at high wind speeds in the revised manuscript.

**III. Wind speed and snow/ice Crystals**

Temperatures mostly below zero and high wind speed rates lead me to the thought in how far ice crystals from local sources may contribute to the measured data set. To me, it seems to be reasonable that, at least, a minor portion of particle concentrations counted, may be ice crystals. Furthermore, crystal structures on particle surfaces may also affect the asymmetry factor (and also sizing) by changing light scattering patterns detected by the Quad-PMT. However, the occurrence of ice crystals depends on the overall inlet system which needs, as stated above, a more detailed description.

> While the aerosol inlet stack is not heated, CASLab is heated to a regular room temperature (~20 to 25 °C), thus we feel it is unlikely that an ice particle would make it to the sensing region of the instrument without melting or evaporating in the sample line between the inlet stack and the instrument.  Furthermore the majority of the air sampled by the WIBS is used as a filtered sheath flow which has a longer residence time in the instrument, effectively heating and drying the air, which would further act to melt/evaporate any ice crystals before they could be detected.  As such we believe the influence of ice crystals on the measurements to be negligible.

**IV. Vessels as potential emission sources**

Even if the marine traffic in this particular area is considered to be rather low, vessels as a potential particle emission source has to be kept in mind though. Attached is a link showing a traffic density map from 2015 (Click on density map button on left):

https://www.marinetraffic.com/en/ais/home/centerx:-59.2/centery:-64.6/zoom:4

As you can see in here, there is a main traffic route in NW direction including mostly tankers, cargo- and fishing vessels. Compared to the back-trajectory analyses in figure 7 (page 14), all wind events (except for E) crossed or brushed the main traffic route for which I think that it has to be considered as a potential emission source to some extent.

> We thank the referee for the useful suggestion and we will include a short discussion of marine vessels as a potential source in the revised manuscript.

V. Geospatial analyses

The data processing of figure 6 (page 12) is unclear to me and needs some further explanation. How were the land class types in combination with back-trajectories processed? Was the trajectory length used? Or was the trajectory "footprint" put onto a, e.g., raster map and blanked out?

> The method to determine time spent over each land class followed three procedural steps:
>
> 1) The land class types were obtained from the sea ice fractional coverage (at 25 km resolution) maps, obtained from the product *Near-Real-Time DMSP SSMIS Daily Polar Gridded Sea Ice Concentrations* (Maslanik, J. and J. Stroeve. 1999), Available from the National Snow and Ice Data Center. In this dataset, sea ice, and land (continent/coast) was marked. Open water was deduced from areas where sea ice <5%. In practice this upper limit could be set to 1% or 10% without impacting the conclusions.
>
> 2) Back trajectory analysis performed using HYSPLIT (Stein et al., 2015); five-day back trajectories (one hour time step) were calculated using the National Centers for Environmental Prediction (NCEP) reanalysis meteorological field. (Stein et al., 2015).
>
> 3) At each time-step (hour) we determine the type of land for the lat/lon point of the back trajectory. We do this for all (hourly) back trajectories. Then the ratio of the occurrences of lat/lon over each type of land divided by the total number of points is derived for the last 12h, 48, 72h, etc.

Technical corrections: Single trajectory plots in figure 7 (page 14) need captions for better allocation.

> We include text in each plot describing the period/wind event it covers to improve clarity in the revised manuscript.

Final comment: The current manuscript provides an interesting data set and will be useful for the whole bioaerosol community and should, therefore, be published. However, the authors need to state the general "case study-nature" of the manuscript more clearly and discuss effects and potential interferences which might occur in this complex environment (e.g. snow and ice, vessels)

more detailed. Furthermore, the inlet system used in this study needs some further description.

We will reiterate the case study nature of the work presented in the final paragraph of the manuscript and we will suggest that further long term studies with accompanying supporting measurements are needed to build up a climatology of bioaerosol events. The other suggestions are dealt with in previous responses to this review.

**References**

Maslanik, J., and J. C. Stroeve (1999), Near-Real-Time DMSP SSMIS Daily Polar Gridded Sea Ice Concentrations, Version 1 [December 2015–January 2016]. Boulder, Colo.: NASA DAAC at the Natl. Snow and Ice Data Cent., doi:10.5067/u8c09dwvx9lm.

Stein, A. F., Draxler, R. R., Rolph, G. D., Stunder, B. J. B., Cohen,M. D., and Ngan, F.: NOAA's HYSPLIT atmospheric transport and dispersion modeling system, B. Am. Meteorol. Soc., 2015, 2059–2077, doi:10.1175/BAMS-D-14-00110.1, 2015

---

## Author Comment (AC3) · 9 Aug 2017

For clarity, the referee's comments are copied in black and our responses are offset in blue.

This manuscript presents measurements of fluorescent aerosol made in Antarctica over a period of weeks using a WIBS. Although the authors find that fluorescent particles are a minor component (a few %) of total aerosol, there are some interesting features in the data worthy of publication. WIBS data is analyzed using a clustering method previously published by this group and 4 component populations are identified. Two of these clusters (together accounting for >97% of the fluorescent aerosol) are only weakly fluorescent and are hypothesized to be non-biological fluorescent aerosol, possibly dust. The other two clusters have more fluorescent intensity, are hypothesized to be biological and one of these is very similar to a cluster identified from laboratory samples of various pollen. Fluorescent loadings are analyzed as a function of wind speed for specific periods of interest and the authors state that high levels of fluorescent aerosols were primarily (though not always) associated with flow from the NE. Back trajectories are also analyzed and the authors posit that fluorescent aerosol (and thus pollen) arrives at the site as a result of long range transport from as far away as South America.

We thank the reviewer for their helpful comments and recommendations which we address below.

**Comments:**

This paper presents the first fluorescent aerosol observations reported for Antarctica and, as such, it is a worthy contribution to the literature. However I found portions confusing and also recommend including more information in certain places. Much of my discomfort arises from the fact that the 5 periods of interest seem rather arbitrary, at least given the information presented. As such, I don't know how to interpret observed differences between these periods or what they mean for fluorescent aerosol in Antarctica more generally. Specific suggestions for improvement are included below.

1. With any discussion of intensity of fluorescent signals the question of calibration arises. I appreciate that there is, as yet, no widely-accepted calibration for fluorescence in the WIBS and it seems that this instrument has been used in numerous laboratory and field studies without significant intentional modification. There is some discussion of this in the discussion of pollen identification (i.e. that the same instrument was used to look at pollen samples and they look very similar in intensity to Cl4) however it would be appropriate to include a more thorough related discussion in the methods section. Do the authors have any information regarding the stability of fluorescent intensity measurements over time? Are the instrument gains used here the same as those used in previously published work from this group? Can you comment on whether or how changes or instability in fluorescent sensitivity would affect the clustering algorithm? Can the authors comment on what kinds of laboratory-generated particles they have observed to fall into the weakly, moderately, medium and highly fluorescent populations? Those categories seem arbitrary and are used only minimally in the subsequent analysis.

Details of the sampling methodology are provided in Ruske et al. (2016). We will include a short description of the methodology in Appendix A where this is discussed.

The instrument used here is periodically sent back to the manufacturer for servicing, where the PMT voltages and xenon powers are noted. No significant changes in these values have been recorded between servicing. In the field prior to the start of measurement, instrument response was checked with fluorescent doped PSLs to verify the instrument is responding sensibly, however, absolute comparison between calibrations is not possible due to variation in fluorescent intensity between batches and the degradation of the doping material with time.

The version of the WIBS used here does not feature multiple gain modes and the detector gain is expected to be similar to that of previous studies.

It is not anticipated that the unsupervised clustering algorithm used here would be sensitive to differences in detector gain/fluorescent instability since absolute values are not referenced to a training dataset as would be the case with supervised methods.

The use of weakly/moderately/highly fluorescent is used as a descriptor to aid the discussion of classification. Generally from our laboratory characterisation we observe pollens to be highly fluorescent; fungal spores to be medium to highly fluorescent; bacteria to be moderate to medium fluorescent and mineral dusts to be weakly fluorescent. Savage et al. (2017) have recently performed a series of systematic laboratory characterisations which demonstrate that these particle types display a comparable broad trend in fluorescent spectra intensities.

2. On a related note, it would be good to include the numbers of particles sampled that fell in each cluster and also the number of particles that saturated the detector. Do the detectors for this WIBS saturate around 2000 counts? If so, given that the stated average intensity in the pollen population is ~1800+/-300 after exclusion of saturating particles, it seems that a substantial fraction of pollen particles would saturate and you might be underestimating the contribution of that population.

We will include the number of particles attributed to each cluster in the revised manuscript.

There is an error in the technical description of the data analysis methods. In this analysis we retain any saturating particles to maximise the PBAP populations. We will correct this error in the revised manuscript and add a short discussion about why we have chosen to retain the saturating particles to maximise PBAP count.

3. A relatively minor point but, in your discussion of asymmetry factor, I believe dust is typically quite fractal (e.g. Bi, Huang et al, ACP 2016 or Yu, Zhu, et al, ACP 2015) yet your dust cluster AF indicates relatively sphere-like. Can the authors provide information to bolster confidence in the retrieved AF from the WIBS? (i.e. any data from calibrations with known aspherical particles or any corroborating reports of relatively spherical dust?)

The simple quadrant detector used here is incapable of detecting such fine detail, which may be captured by a more sophisticated detector such as the dual CMOS array used in the MBS for example. From our own laboratory characterisation experiments we have found that mineral dusts exhibit asymmetry factors of around 10, however, this work has not been published. Savage at al., (2017) performed a systematic characterisation of many particle types of interest using a WIBS-4A, which features a similar quadrant detector to the instrument used in this study. They characterised 13 mineral dusts of which many had an AF of approximately 10.

The quadrant detector AF proxy was calibrated for rod like particles using elliptical haematite as described in Kaye et al., (2007)

4. I am confused by the discussion surrounding the wind events. First, the authors define a level above which they consider fluorescent concentrations elevated and imply that they are going to look at periods where that happened. Then, however, two of the five periods in table 2 don't have elevated fluorescent concentrations (the 2nd and the 4th) while there are periods that seem to have elevated fluorescent concentrations that are not included in the analysis (i.e. early on in the project and on 11/29). Is the selection driven mainly by wind speed and direction? Why include the 5th period and not periods from 20-21 and 29 Nov? Are these just meant to be case studies of the different combinations of wind and aerosol loadings observed? Please clarify how these 5 periods of interest were chosen. It would also be helpful if these periods were marked in Figure 2 so that the reader doesn't have to mentally combine the table and the figure.

First we chose wind event A, based on its high concentrations of fluorescent material and PBAP cluster as a period of significant interest. This period featured high wind speeds from the NE, which is characteristic of the site (e.g., Renfrew & Anderson, 2002; Van Lipzig, et al., 2004) and confirmed by our own meteorological measurements during the experiment, as shown below:

[Figure]

**Figure 1. Polar histogram of wind speed and direction during MAC measurement period. Frequency indicates the number of 5 minute integrations. Rings indicate 5ms$^{-1}$ wind speed intervals.**

Wind events B, C & D were chosen for comparison to event A as they have similar speeds and directions, yet the fluorescent and PBAP concentrations were significantly less than for event A. Event E was chosen as a case study demonstrating the much less frequent SW. We will clarify the selection criteria in the revised manuscript and include a shaded area highlighting the events in the middle and bottom panels of figure 2 as requested.

5. In the text, the authors state that high levels of fluorescent aerosols were primarily associated with flow from the NE but I don't think this statement is supported by the data presented. To me it seems that there was one period of fluorescent enhancement from the NE and one from the W. There are possibly even two instances of high loadings with westerly flow if you consider the noisy but relatively elevated concentrations at the beginning of the project in addition to what was seen in the 5th highlighted period. Other instances of flow from either direction don't necessarily bring

elevated concentrations and I don't know what the explanation is for this behavior but I don't think it's as clean as currently presented.

We will rephrase this to state that while there are both wind events featuring high fluorescent concentrations from the NE and SW, only the NE wind event A features any significant PBAP cluster concentrations.

6. The authors also state that they see enhancements in the ratio of fluorescent to total aerosol at particular times. It is nearly impossible to assess this ratio from the graph presented. I recommend adding a panel or a figure to show a time series of the fluorescent fraction, possibly showing two traces where one shows the "dust-like" fluorescent fraction and one shows the PBAP fluorescent fraction.

We agree that is difficult to determine the fluorescent and PBAP to total aerosol concentration ratio from the figures presented in the manuscript. Showing the ratio time series as a panel in figure 2 made the figure too busy so it was omitted. We feel the best way to show the ratios are as a polar plot to demonstrate the influence of wind speed and direction, which we provide below and will include in the revised manuscript.

[Figure]

Figure 2. Polar plot of the ratio of fluorescent (left panel) and PBAP (right panel) to total aerosol concentration. Polar plots are a function of wind speed and wind direction, with concentric rings representing 5 ms$^{-1}$ increments.

7. I don't fully understand Figure 3. Was this made from the average of all periods when the wind was from the NE and, if so, how was this average calculated? How is it that the plot for total fluorescent particles has a component in the SW quadrant but the other two do not? In panel b, it is labelled as dust but also as Cl1. I thought dust was Cl3 and Cl1 was unclassified. Either way, why show the plot for one but not the other?

The presented figures are for the time period specified for wind event A to examine the influence of wind speed and direction, demonstrating a "hot spot" ENE at wind speeds > 10 ms$^{-1}$. We accept that the period being examined is not clear and we will clarify this in the revised manuscript. We will also revise the labelling of Cl1 to unclassified in the text to be consistent with table 1. We chose to show Cl1 due to it similar wind response to PBAP.

8. In the caption of Fig 4 it is stated that these plots are only for the NE wind event with the highest fluorescent loadings however the text on lines 1-3 of the same page implies that it is for all of the selected events. Please make these consistent. If the graphs are really only for a single event, it would be interesting to know whether similar behaviour was observed during other periods. What does it look like if similar graphs are made for the westerly event that had relatively high fluorescent loadings?

The figure caption displays this correct period (wind event A).  We will correct the text to state this.

The SW wind event (E) does display an increase in the total fluorescent concentration with increasing wind speed, however, very little of Cl1 is observed and virtually no PBAP.  The fluorescent ratio is also constant with increasing wind speed during this event.

[Figure]

**Figure 3.  Fluorescent particle concentrations as a function of wind speed for the period 14/12/2015 - 16/12/2015 for: (a) Total fluorescent particles, $N_{FL}$; (b) Moderately fluorescent particles, $N_{Cl1}$; (c) Ratio of total fluorescent particles to total particle concentration $N_{FL}:N_{TOT}$ and (d)$N_{PBAP}(N_{Cl2}+N_{Cl4})$**

9. I am not well-versed in calculations of flux, and I cannot speak to the validity of the method used here. In any case, I don't really see the point of calculating a flux under the present circumstances. If the elevated concentrations are episodic and not systematically associated with a particular flow direction or meteorological context, then it doesn't seem that this is likely to represent flux from sea ice or the ocean or any other dispersed source but, rather, will represent flux from a particular but unidentified bioaerosol source at an unknown location and I don't see the utility. Flux from the local environment might be better assessed by looking at wind events without elevated fluorescent concentrations but, again, I don't know enough about flux to know if this would be robust or even possible.

Previous studies have used simple concentration enhancements as a function of wind speed to imply local emissions and emission fluxes from surfaces (e.g., Sesartic and Dallafior, 2011, and references therein). We show that such approaches are overly simplistic and more robust micrometeorological methods will be needed for bioparticle flux estimates, particularly in these ice dominated ecosystems.

10. With regard to the airmass trajectory analysis, it would be nice to see maps for all of the events discussed. Was event A the only time that flow arrived from S. America or was there a time with similar back trajectories but little fluorescent aerosol enhancement?

We selected the airmass trajectories to display events of interest for comparison. We will amend figure 7 to include a representative trajectory from each event. There was only one other period displaying significant flow from S. America (27/11/15, shown below), however, this coincided with some of the lowest fluorescent concentrations observed, with no PBAP detected.

[Figure]

**Figure 4. 5-day back trajectory analysis using the NAME particle dispersion model for 27/11/15.**

11. As stated above, much of my discomfort with this paper arises from the fact that so much of the discussion centers on analysis of 5 events (and of those 5, only one or two get much attention) and the selection of these events is unclear to me. It is therefore difficult to develop a sense for how representative they might be, how to interpret the variability between them or what they mean in a larger context. The text is often written as though systematic relationships have been found which I find a bit misleading given that the study duration was relatively short and these "relationships" are extrapolated from single events. I recommend rephrasing these statements and revisiting the data analysis to more clearly delineate the observations themselves, the generalizations made based on the observations and the limitations to these generalizations imposed by the short duration of the study and the episodic nature of the environment.

It is not our intention to present the measurements from this short, opportunistic pilot study to be generally representative of Antarctic bioaerosol. As we replied to referee #2, we will reiterate the case study nature of the work presented in the final paragraph of the manuscript and we will suggest that further long term studies with accompanying supporting measurements are needed to build up a climatology of bioaerosol events to assess the influence of long range transport of PBAP/pollen from South America.

**References**

Kaye, P. H., Aptowicz, K., Chang, R. K., Foot, V., and Videen, G.: Angularly Resolved Elastic Scattering from Airborne Particles, Opt. Biol. Part., 31–61, 2007.

Renfrew, I. A. and Anderson, P. S. (2002), The surface climatology of an ordinary katabatic wind regime in Coats Land, Antarctica. Tellus A, 54: 463–484. doi:10.1034/j.1600-0870.2002.201397.x

Ruske, S., Topping, D. O., Foot, V. E., Kaye, P. H., Stanley, W. R., Crawford, I., Morse, A. P., and Gallagher, M. W.: Evaluation of machine learning algorithms for classification of primary biological aerosol using a new UV-LIF spectrometer, Atmos. Meas. Tech., 10, 695-708, https://doi.org/10.5194/amt-10-695-2017, 2017.

Savage, N., Krentz, C., Könemann, T., Han, T. T., Mainelis, G., Pöhlker, C., and Huffman, J. A.: Systematic Characterization and Fluorescence Threshold Strategies for the Wideband Integrated Bioaerosol Sensor (WIBS) Using Size-Resolved Biological and Interfering Particles, Atmos. Meas. Tech. Discuss., https://doi.org/10.5194/amt-2017-170, in review, 2017.

Sesartic, A. and Dallafior, T. N.: Global fungal spore emissions, review and synthesis of literature data, Biogeosciences, 8, 1181-1192, https://doi.org/10.5194/bg-8-1181-2011, 2011.

van Lipzig, N. P. M., Turner, J., Colwell, S. R. and van Den Broeke, M. R. (2004), The near-surface wind field over the Antarctic continent. Int. J. Climatol., 24: 1973–1982. doi:10.1002/joc.1090

---

## Author Response (AR2)

For clarity, the co-author's comments are copied in black and our responses are offset in blue. We thank the co-author for their helpful comments and recommendations which we address below.

Comments to the Author:

Authors,

Thank you for the responses you posted to each of the referee comments and for the revised version. After reading comments following the second round of review, I agree with both referees that the edits are not yet sufficient to accept the manuscript for publication. A number of areas remain where clarification and revision of the text can improve the manuscript. In general, both suggested that responses to their comments need to be translated more specifically into a revised version rather than only in the response document. Additionally, in several cases I agree with the last round of referee comments that the results need to be more carefully placed in appropriate context of their applicability to understand Antarctic aerosol in general. I encourage you to read through both the original set of referee comments and the most recent round to make specific improvements to the manuscript. If there are a minority of comments that you disagree with, please justify those cases individually. I anticipate that the manuscript will be acceptable once the text is sufficiently revised.

Regards,

Alex Huffman

We thank the co-editor for their helpful comments and suggestions.

Non-public comments to the Author:

I wanted to share more specific details in the non-public section. Referee #3 published publically available comments that you should be able to see. Referee #1 submitted only confidential comments. While s/he rated the manuscript as acceptable after technical corrections, it is clear in her/his comments that s/he did not consider the responses to be sufficient. For example s/he stated that the authors were "somewhat dismissive" of original comments, including some other colorful comments, and s/he strongly encouraged me to suggest that the comments in the review document be processed into a revised text. In particular s/he suggested that a brief discussion be included on the limitations of using an instrument that has no quantifiable fluorescence calibration. You and I obviously know that doing the experimental work to make this a reality is not trivial, and having done it before your field measurements is impossible now. I don't disagree that including a few sentences of text on the issue is a reasonable and manageable request, however.

We will include a discussion of a lack of an available fluorescence calibration standard at the time of deployment at the end of section 2.2 as requested.

So I suggest you go back to both rounds of review by these referees and relatively carefully include manuscript revisions. To help you ignore certain comments that I thought were finished or superfluous, I went through the original comments by Referee #1, with the following brief responses:

Point 1

- Regarding manuscript title. I don't disagree with their comment here that including "fluorescence" in the name would benefit the reader. I don't feel strongly that the word "bioparticles" be removed in favor of "fluorescent," but at least including the latter brings some clarity and benefit.

We agree to revise the tile of the revised manuscript to:

"Real Time Detection of Airborne Fluorescent Bioparticles in Antarctica"

- P4L22: I'm okay with a non-response here. Confusing comment.

- P4L30: Here is a place that including extra discussion and references could be useful.

We will include the following at the end of P5L1 and split the remainder into a new paragraph:

"PBAP of interest (e.g., pollen, bacteria & fungal spores) have been demonstrated to show a detectable autofluorescent response with the WIBS (Hernandez et al., 2016, Savage et al., 2017). Non-fluorescent particles will exhibit fluorescent signal below the instrument fluorescence threshold, thus the fluorescent signal will be clipped at zero in the processed data as described in Crawford et al., (2015), however, this information and the particle size is still recorded and used to define the non-fluorescent particle population. Non-fluorescent particles are by default classified as non-biological by this technique."

- P5L16: Include some aspect of your response in the revision.

We will include the following as a new paragraph at P5L20:

"The laboratory categorisation and classification of bioaerosols of interest is an ongoing area of research. To date there have been two significant systematic laboratory characterisation studies published using a similar instrument (WIBS-4A); Hernandez et al., (2016) and Savage et al., (2017). We have also performed our own characterisation for the purpose of validating machine learning algorithms experiments (e.g., Ruske et al., 2017 & Crawford et al., 2015). The Hernandez et al., (2016) study characterised the autofluorescence of 14 bacterial, 13 pollen and 29 fungal spore samples. The Savage et al., (2017) study characterised 3 bacterial, 5 fungal, 14 pollen, 12 pure biofluorophore, 13 mineral dust, 6 HULIS, 3 PAH, 7 combustion soot and smoke, 3 brown carbon and 3 miscellaneous non-biological particle samples. These studies showed that each particle type demonstrated a broad characteristic autofluorescence, size and asymmetry factor that can be used to interpret and classify ambient measurements, e.g., bacteria were found to predominantly fluoresce in channel FL1 and were generally under 2.5 µm in diameter. While these studies are not exhaustive, the authors note that the fluorescent spectra observed should hold as a broad trend for each particle type.

We use such libraries to aide interpretation of our results, along with our own laboratory measurements (provided in Appendix A)."

Point 2, P4L1: Ok

Point 3

- First point, Ok

- P4L22: Include response in text.

We will revise P4L22 to the following for clarity:

"The instrument has an inlet flow of 2.35 L min$^{-1}$, the majority of which is filtered with a HEPA filter to remove all particles, such that the 0.23 L min$^{-1}$ sample flow is sheathed in particle free air to constrain the aerosol into a controlled jet and to minimise contamination of the optics.

- P5L3: Include in text.

We have included the pertinent details of our response in the text we will add as a response to P5L16, as shown in a previous response.

- P5L1: Ok.

- P16L1: Include in text.

We will include the requested text in the revised manuscript.

Point 4: Ok

Point 5: Ok

With respect to Referee #3, respond carefully to the second round of review. Additionally:

Point 1: Similar to comments from Referee #1, include a more thorough response to questions that the referee brings up here. Also include some aspect of your response into the manuscript.

We acknowledge that we misinterpreted the referee's request to provide details in the discussion to be related to the pollen characterisation experiment, and not a discussion of fluorescent calibration as was intended. We apologise and will rectify this error. We will include a discussion on the lack of a fluorescent calibration standard at the time of deployment and how this may influence the subsequent analysis in the revised manuscript.

We will insert the following text at end of section 2.2:

"At the time of deployment no robust fluorescence calibration method existed for UV-LIF spectrometers. Since this time the first successful calibration methods for WIBS type instruments have become available (Robinson et al., 2017). While the data presented here is uncalibrated, the instrument was routinely sent back to the

manufacturer for servicing where no significant changes in the PMT voltages and gains or xenon flash lamp powers were found.  At the start of the of measurement period the  instrument response was checked with fluorescent doped polystyrene latex spheres to verify the instrument was responding sensibly, however, absolute comparison between calibrations is not possible due to variation in fluorescent intensity between  batches  of  particles and  the  degradation  of  their  doping material  with time.  As the unsupervised learning method employed in this study requires no *a priori* information, the lack of calibration should not impact the analysis as the method groups similar data points together for subsequent analysis. The details of this method are described in the next section.“

Point 3: The point that the referee raises about AF here is a good one that s/he brings up again in the second round. See those follow-up comments.

In addition to our response to the second round we will also include a short discussion of observed AF values for mineral dusts when discussing the classification of cluster Cl1, both from our previous measurements from INUPIAQ during Saharan dust events and the recent Savage at al., (2017) characterisation experiments.

We revise P6L22 to the following:

"Particles in this cluster were small, $D_p$ ~1.3 µm, with an AF value of ~11, suggesting near spheroidal particles.  The AF value reported here is consistent with previous WIBS measurements at an Alpine mountain top site during a Saharan dust event, where a distinct dust cluster was observed with an average AF of ~7 (Crawford et al., 2016).   It is also consistent with several mineral dust samples which were systematically sampled as part of a larger fluorescence characterisation experiment and which displayed average AF values of 10. (Savage et al., 2017).  Additionally, this characterisation experiment showed the majority of PBAP samples to display AF values significantly greater than 10."

Point 11: This is also an important point that the referee comes back to in round 2. I would suggest also include text in the abstract quickly but appropriately highlighting the context through which the study should present Antarctic aerosol.

We will reiterate the short, case study nature of this work in the abstract and we will also include the possibility or marine traffic as a source of PBAP in the abstract.  We will revise P1L15 to the following:

"Likely distal sources identified by back trajectory analyses and dispersion modelling were the coastal ice margin zones in Halley Bay consisting of bird colonies with likely associated high bacterial activity together with contributions from exposed ice margin bacterial colonies but also long range transport from the southern coasts of Argentina  and  Chile.   Dispersion modelling also demonstrated emissions from shipping lanes and as such, marine anthropogenic sources cannot be ruled out."

We will also revise P1L20 to the following:

> "While this short pilot study is not intended to be generally representative of Antarctic aerosol, it demonstrates the usefulness of the UV-LIF measurement technique for the quantification of Antarctic airborne bioaerosol concentrations, and to understand their dispersion. The potential importance for microbial colonisation of Antarctica is highlighted."

Additional general and typographical comments (line numbers refer to revised/tracked manuscript text):

P1L10: "Clearly" seems overselling here.

> We will revise this to "likely".

P4, First paragraph of Section 2.2: The references to previous WIBS work seems overly self-citatory here. While I recognize the critical contributions your group has made, it would be useful to include a slightly broader perspective here.

> We will include additional references in the revised manuscript as requested.

P4, L26: AF is not rigorously defined and only mentioned as a 'shape factor' which is a stretch.

> We will revise this to the following:

> > "Aerosol in the sample flow is illuminated by a 635nm laser and the resultant scattered light is used to determine the particle size and a proxy for particle shape using a quadrant detector, asymmetry factor (AF) and is interpreted as follows: AF < 10-15 is indicative of near spherical particles, AF > 20 aspherical particles, and AF > 30 fibre or rod like particles, where laboratory characterisations using corn starch flour to represent irregular particles and ellipsoidal haematite particles were used as an analogue for rod-like bacterial particles to determine these thresholds (Kaye et al. 2007)."

P4, L34: Missing closing parentheses behind 420-650 nm.

> We will correct this in the revised manuscript.

P5, L6: I don't think you have yet defined PBAP

> We will define PBAP here.

P5, L12: sigma is not yet defined

> We will explicitly state this as 3 standard deviations for clarity.

P6, L15: More clearly define what you mean by "identified" here.

> We will revise this to:

> *"A subset of approximately 17,000 particles were identified as exhibiting fluorescent intensities greater than the fluorescence threshold…"*

P6, L25: "Significantly more aspherical" This relates to the comments by Referee #3 about AF.

> *We will remove "significantly".*

P8, L2: You combined clusters 2 and 4 into an assumed "PBAP" cluster, which is fine for analysis. The terminology applied here implies lesser uncertainty than I think is appropriate, however. I strongly suggest calling this something like "PBAP cluster" etc. so that it is easy to distinguish between "PBAP" as directly measured and "PBAP cluster" from those particles assigned to the name after clustering of fluorescent particles.

> *We will revise this as suggested.*

P12, L4: I agree with the authors that including this first-approximation of flux values is worthwhile here, but I think it is reasonable to heed the referee comments here by at least putting the magnitude of bounds on the uncertainty. I.e. are you talking about order of magnitude uncertainty? Plus/minus 20%, off by 3 order of magnitude, etc.

> *We will revise the flux estimate section of the paper after revisiting referee #3's comments about their applicability and purpose they serve. This was meant as cautionary tale for previous flux estimates using this approach. We will include error bounds on the presented values using the standard deviations of the observed number concentrations.*

P14, L20: Word "proven" is a bit strong here.

> *We will remove "proven" in the revised manuscript.*

P16, L1: ",based on …. comparison with … laboratory studies."

> *We will revise this as suggested.*

P16, L32: "Four typical pollens" … do you mean that they were all aerosolized together? Please clarify.

> *The samples were all aerosolised separately. We will revise the previous sentence to the following to clarify this:*

>> *"Sample particles of interest were separately aerosolised into a large, clean HEPA filtered containment chamber (incorporating a recirculation fan), from which the WIBS-3D drew measurement samples. Dry materials were aerosolised directly from small quantities of powder using a filtered compressed air jet. The sample chamber was cleaned with absorbent paper and sodium hypochlorite in between samples to minimise contamination (Ruske et al., 2017)."*

P17, Table A1: If Cl2 is likely to be pollen, what is Cl1? Please clarify

> *Cluster Cl1 is unidentified but may be caused by fragmentation of the pollen samples during aerosolisation. We will include this after the discussion of Cl2 in the revised manuscript.*

**Referee #3 Report**

The authors have clarified their selection of time periods to analyze, however their responses to other comments by myself and other reviewers are rather minimal and unsatisfying (i.e. I don't see where they have amended anything in response to reviewer 2's questions about the back trajectory analysis).

Additionally, now that the authors have included the number of particles observed in each cluster, I wonder if this might be a bit of over-analysis given the low statistics. Especially given that it seems like a single anthropogenic source from a ship could be responsible for quite a lot of what seemed to be PBAP. Even before this addition, multiple reviewers had commented about the robustness and applicability of these measurements and, in response, the authors have added one sentence at the end calling for more work in the future. In my opinion a larger and more thoughtful re-write would have been appropriate.

> We feel that the low counts of PBAP observed here serves to highlight the utility of the UV-LIF technique in environments where low numbers of particles of interest may be masked by several orders of magnitude more of common aerosol (e.g., sea salt) when using traditional collection methods, such as impaction on filters. We disagree that the other referees have called into question the applicability and robustness of the measurements in their assessment of the manuscript, however, we will revise our conclusions to include the possibility that the PBAP observed during wind event A may also be due to anthropogenic emissions from marine traffic. However, based on previous measurements of anthropogenic aerosol emissions this is unlikely
>
> We will revise P16L17 in the summary & conclusions to the following:
>
> > "The presence of particles characteristic of pollen is evidence towards the latter conclusion, however, the dispersion model results also display emissions from regions which feature significant marine traffic, and thus anthropogenic marine sources for these observations cannot be ruled out."
>
> During the review phase of this manuscript, the Savage et al., (2017) WIBS bioaerosol characterisation paper was accepted into public discussion. As part of this paper the authors investigated the effect of increasing the fluorescence threshold from the forced trigger mean + 3 standard deviations to the mean + 9 standard deviations in an attempt to minimise the inclusion of non-biological particles into the fluorescent population. They found that generally that all PBAP of interest would fluoresce well above this increased threshold, while non-biological interferents would not, effectively equating the fluorescent concentration to that of the true PBAP concentration. We have briefly investigated the effect of using this increased threshold in our analysis; this has the effect for greatly reducing the fluorescent number concentration to typically less than a few per litre, with peaks of 5 to 20 $L^{-1}$ during wind event A. The cluster analysis on this reduced input sample features 3 highly fluorescent clusters and 2 medium fluorescence clusters, relative to the new $9\sigma$ threshold. It was seen that using the increased threshold removes the likely non-biological material, leaving only

any significant PBAP concentrations during wind event A. This is consistent with the results we present using the 3σ threshold. The removal of weakly fluorescent non-biological aerosol from the clustering input improves the PBAP counting statistics to ~1000 particles observed during wind event A, as PBAP are not misattributed into the non-biological centroids as is possible in the 3σ threshold case.

I am still uncomfortable with the flux calculation as presented though I suppose the main conclusion is just that the local source, if there is one, is tiny. In response to my comment the authors say that they show that previous methods are overly simplistic but that's not at all how I read that section of the paper. To me it seems like the authors take these very simplistic tools and apply them to a scenario where it's not clear that they apply or give useful information but present it as the best that can be done.

The unconstrained, single height flux calculation presented is highly uncertain, as we state in the manuscript. After discussion of its scientific value, we chose to include the flux in the manuscript to highlight issues with this technique and raise questions about its applicability, since this method has been employed in frequently cited modelling studies. Given the uncertainty of the method and the lack of other estimates to compare our values to, we feel it is best to revise the flux section to firmly state the uncertainty in the presented values and to advise caution in interpreting a result which may not be applicable to the circumstances of the measurements. The point here is that previous similar approaches have been used to estimate bioaerosol fluxes for model assimilation and such approaches can, as highlighted by the discussions, be problematic. We revise this section to the following:

"In the past, short term wind driven enhancements in number concentrations have been used to infer the existence of local surface sources, e.g., Sesartic and Dallafior (2011), however, deriving an aerosol flux from single height concentration measurements can lead to highly uncertain results, Petelski and Piskozub (2006) & Pryor et al., (2008). If we assume that the majority of the larger fluorescent particles (clusters Cl1 and PBAP cluster) are locally re-suspended then a net flux for these could be estimated using the approach of Sesartic and Dallafior (2011), resulting in fluxes for wind event A of $F_{Cl1}$ = 7.2 ± 11.8, and $F_{PBAP}$ = 1.1 ± 2.6 $m^2 s^{-1}$. However such calculations based on these crude assumptions are very uncertain.

Air mass back trajectory analysis and dispersion modelling (next section) reveal that the aerosol observed during wind event A may be from distal sources, and as such, the local surface source flux estimates presented are invalid in this case. These values are presented to highlight the difficulty in estimating bioaerosol fluxes using these methods, which have previously been used in model assimilations, and they should not be over-interpreted given their uncertainty and may be invalid in this case."

I also think there could be much more thoughtful additions regarding the variable performance of the clustering algorithm and the impacts of that on the conclusions. For example, if the asymmetry factor cannot distinguish the "fine detail" between biological particles and dust (which should actually look very different from each other) then I think it is wrong to try to interpret small

variations in that measurement as indicative of different clusters. What happens if that piece of information is omitted?

In our prior response to question 3 by referee #3 we state that the quadrant detector is incapable of resolving fine detail. By this we meant that it cannot detect surface morphology as is possible with a dual CMOS array (as in the Ruske et al., 2017 study), however, differentiation between spheroidal, ellipsoidal and rod like particles is possible which can provide a useful piece information alongside other data when interpreting cluster results. Certainly in the case of clusters 1 & 3 we feel the difference in AF values is sufficiently large to be able to separate the two when also considering their fluorescence and size values, whereas the much more fluorescent clusters can clearly be distinguished based on their fluorescent intensities and size.

As requested we have removed AF from the clustering to demonstrate its effect. This tends to have the effect of promoting fragmentation of the clusters into subclusters based on our previous research into HCA algorithms, however, this detail was omitted from Crawford et al., (2015) for the sake of brevity. We see in this case that the removal of AF results in cluster fragmentation; the previous 4 cluster solution with AF included is fragmented into 12 clusters when AF is omitted from the clustering input, which we display visually below.

[Figure]

**Figure 1. Cluster solutions with AF included.**

[Figure]

**Figure 2. Cluster solutions with AF included.**

Here it can be seen that the original cluster 3 (Fig. 2) has been split into clusters 4 and 12 when AF is omitted (Fig. 3) on the basis of their size; original cluster 1 has been split into clusters 3 and 10 for the same reason; the remaining moderately to highly fluorescent fragmented clusters by in large compare with original PBAP clusters 2 & 4, the fragmentation of which is largely depended on the choice of HCA linkage used. Given the low numbers of particles assigned to each of these fragmented PBAP clusters (most contain single digit counts) we would still combine them into one larger overall PBAB cluster in our analysis.

In general, though the authors have at least responded to some of the larger comments, it still feels like the dataset is a bit oversold and the conclusions are simply not completely convincing or particularly interesting.

It was our intention to demonstrate the value of using UV-LIF spectrometers to detect very low counts of PBAP in a climate sensitive region, which may be missed by traditional methods, rather than to present the results as being generally representative of Antarctic bioaerosol. We feel that this dataset shows that PBAP may be transported from South America, as hypothesised by Pearce et al., (2009), and is of interest to the community for this reason. We accept that other sources such as marine traffic may also be the source of the observed PBAP and this will be clarified in the revised manuscript as detailed in an earlier response. We are encouraged however that the observations have generated new discussions.

For clarity, the co-author's comments are copied in black and our responses are offset in blue. We thank the co-author for their helpful comments and recommendations which we address below.

Comments to the Author:

Authors,

Thank you for the responses you posted to each of the referee comments and for the revised version. After reading comments following the second round of review, I agree with both referees that the edits are not yet sufficient to accept the manuscript for publication. A number of areas remain where clarification and revision of the text can improve the manuscript. In general, both suggested that responses to their comments need to be translated more specifically into a revised version rather than only in the response document. Additionally, in several cases I agree with the last round of referee comments that the results need to be more carefully placed in appropriate context of their applicability to understand Antarctic aerosol in general. I encourage you to read through both the original set of referee comments and the most recent round to make specific improvements to the manuscript. If there are a minority of comments that you disagree with, please justify those cases individually. I anticipate that the manuscript will be acceptable once the text is sufficiently revised.

Regards,

Alex Huffman

Non-public comments to the Author:

I wanted to share more specific details in the non-public section. Referee #3 published publically available comments that you should be able to see. Referee #1 submitted only confidential comments. While s/he rated the manuscript as acceptable after technical corrections, it is clear in her/his comments that s/he did not consider the responses to be sufficient. For example s/he stated that the authors were "somewhat dismissive" of original comments, including some other colorful comments, and s/he strongly encouraged me to suggest that the comments in the review document be processed into a revised text. In particular s/he suggested that a brief discussion be included on the limitations of using an instrument that has no quantifiable fluorescence calibration. You and I obviously know that doing the experimental work to make this a reality is not trivial, and having done it before your field measurements is impossible now. I don't disagree that including a few sentences of text on the issue is a reasonable and manageable request, however.

> We will include a discussion of a lack of an available fluorescence calibration standard at the time of deployment at the end of section 2.2 as requested.

So I suggest you go back to both rounds of review by these referees and relatively carefully include manuscript revisions. To help you ignore certain comments that I thought were finished or superfluous, I went through the original comments by Referee #1, with the following brief responses:

Point 1

- Regarding manuscript title. I don't disagree with their comment here that including "fluorescence" in the name would benefit the reader. I don't feel strongly that the word "bioparticles" be removed in favor of "fluorescent," but at least including the latter brings some clarity and benefit.

We agree to revise the tile of the revised manuscript to:

"Real Time Detection of Airborne Fluorescent Bioparticles in Antarctica"

- P4L22: I'm okay with a non-response here. Confusing comment.

- P4L30: Here is a place that including extra discussion and references could be useful.

We will include the following at the end of P5L1 and split the remainder into a new paragraph:

"PBAP of interest (e.g., pollen, bacteria & fungal spores) have been demonstrated to show a detectable autofluorescent response with the WIBS (Hernandez et al., 2016, Savage et al., 2017). Non-fluorescent particles will exhibit fluorescent signal below the instrument fluorescence threshold, thus the fluorescent signal will be clipped at zero in the processed data as described in Crawford et al., (2015), however, this information and the particle size is still recorded and used to define the non-fluorescent particle population. Non-fluorescent particles are by default classified as non-biological by this technique."

- P5L16: Include some aspect of your response in the revision.

We will include the following as a new paragraph at P5L20:

"The laboratory categorisation and classification of bioaerosols of interest is an ongoing area of research. To date there have been two significant systematic laboratory characterisation studies published using a similar instrument (WIBS-4A); Hernandez et al., (2016) and Savage et al., (2017). We have also performed our own characterisation for the purpose of validating machine learning algorithms experiments (e.g., Ruske et al., 2017 & Crawford et al., 2015). The Hernandez et al., (2016) study characterised the autofluorescence of 14 bacterial, 13 pollen and 29 fungal spore samples. The Savage et al., (2017) study characterised 3 bacterial, 5 fungal, 14 pollen, 12 pure biofluorophore, 13 mineral dust, 6 HULIS, 3 PAH, 7 combustion soot and smoke, 3 brown carbon and 3 miscellaneous non-biological particle samples. These studies showed that each particle type demonstrated a broad characteristic autofluorescence, size and asymmetry factor that can be used to interpret and classify ambient measurements, e.g., bacteria were found to predominantly fluoresce in channel FL1 and were generally under 2.5 μm in diameter. While these studies are not exhaustive, the authors note that the fluorescent spectra observed should hold as a broad trend for each particle type.

> We use such libraries to aide interpretation of our results, along with our own laboratory measurements (provided in Appendix A)."

Point 2, P4L1: Ok

Point 3

- First point, Ok

- P4L22: Include response in text.

> We will revise P4L22 to the following for clarity:

> > "The instrument has an inlet flow of 2.35 L min$^{-1}$, the majority of which is filtered with a HEPA filter to remove all particles, such that the 0.23 L min$^{-1}$ sample flow is sheathed in particle free air to constrain the aerosol into a controlled jet and to minimise contamination of the optics.

- P5L3: Include in text.

> We have included the pertinent details of our response in the text we will add as a response to P5L16, as shown in a previous response.

- P5L1: Ok.

- P16L1: Include in text.

> We will include the requested text in the revised manuscript.

Point 4: Ok

Point 5: Ok

With respect to Referee #3, respond carefully to the second round of review. Additionally:

Point 1: Similar to comments from Referee #1, include a more thorough response to questions that the referee brings up here. Also include some aspect of your response into the manuscript.

> We acknowledge that we misinterpreted the referee's request to provide details in the discussion to be related to the pollen characterisation experiment, and not a discussion of fluorescent calibration as was intended. We apologise and will rectify this error. We will include a discussion on the lack of a fluorescent calibration standard at the time of deployment and how this may influence the subsequent analysis in the revised manuscript.

> We will insert the following text at end of section 2.2:

> > "At the time of deployment no robust fluorescence calibration method existed for UV-LIF spectrometers. Since this time the first successful calibration methods for WIBS type instruments have become available (Robinson et al., 2017). While the data presented here is uncalibrated, the instrument was routinely sent back to the

> manufacturer for servicing where no significant changes in the PMT voltages and gains or xenon flash lamp powers were found. At the start of the of measurement period the instrument response was checked with fluorescent doped polystyrene latex spheres to verify the instrument was responding sensibly, however, absolute comparison between calibrations is not possible due to variation in fluorescent intensity between batches of particles and the degradation of their doping material with time. As the unsupervised learning method employed in this study requires no *a priori* information, the lack of calibration should not impact the analysis as the method groups similar data points together for subsequent analysis. The details of this method are described in the next section."

Point 3: The point that the referee raises about AF here is a good one that s/he brings up again in the second round. See those follow-up comments.

> In addition to our response to the second round we will also include a short discussion of observed AF values for mineral dusts when discussing the classification of cluster Cl1, both from our previous measurements from INUPIAQ during Saharan dust events and the recent Savage at al., (2017) characterisation experiments.

> We revise P6L22 to the following:

>> "Particles in this cluster were small, $D_p$ ~1.3 μm, with an AF value of ~11, suggesting near spheroidal particles. The AF value reported here is consistent with previous WIBS measurements at an Alpine mountain top site during a Saharan dust event, where a distinct dust cluster was observed with an average AF of ~7 (Crawford et al., 2016). It is also consistent with several mineral dust samples which were systematically sampled as part of a larger fluorescence characterisation experiment and which displayed average AF values of 10. (Savage et al., 2017). Additionally, this characterisation experiment showed the majority of PBAP samples to display AF values significantly greater than 10."

Point 11: This is also an important point that the referee comes back to in round 2. I would suggest also include text in the abstract quickly but appropriately highlighting the context through which the study should present Antarctic aerosol.

> We will reiterate the short, case study nature of this work in the abstract and we will also include the possibility or marine traffic as a source of PBAP in the abstract. We will revise P1L15 to the following:

>> "Likely distal sources identified by back trajectory analyses and dispersion modelling were the coastal ice margin zones in Halley Bay consisting of bird colonies with likely associated high bacterial activity together with contributions from exposed ice margin bacterial colonies but also long range transport from the southern coasts of Argentina and Chile. Dispersion modelling also demonstrated emissions from shipping lanes and as such, marine anthropogenic sources cannot be ruled out."

We will also revise P1L20 to the following:

"While this short pilot study is not intended to be generally representative of Antarctic aerosol, it demonstrates the usefulness of the UV-LIF measurement technique for the quantification of Antarctic airborne bioaerosol concentrations, and to understand their dispersion. The potential importance for microbial colonisation of Antarctica is highlighted."

Additional general and typographical comments (line numbers refer to revised/tracked manuscript text):

P1L10: "Clearly" seems overselling here.

We will revise this to "likely".

P4, First paragraph of Section 2.2: The references to previous WIBS work seems overly self-citatory here. While I recognize the critical contributions your group has made, it would be useful to include a slightly broader perspective here.

We will include additional references in the revised manuscript as requested.

P4, L26: AF is not rigorously defined and only mentioned as a 'shape factor' which is a stretch.

We will revise this to the following:

"Aerosol in the sample flow is illuminated by a 635nm laser and the resultant scattered light is used to determine the particle size and a proxy for particle shape using a quadrant detector, asymmetry factor (AF) and is interpreted as follows: AF < 10-15 is indicative of near spherical particles, AF > 20 aspherical particles, and AF > 30 fibre or rod like particles, where laboratory characterisations using corn starch flour to represent irregular particles and ellipsoidal haematite particles were used as an analogue for rod-like bacterial particles to determine these thresholds (Kaye et al. 2007)."

P4, L34: Missing closing parentheses behind 420-650 nm.

We will correct this in the revised manuscript.

P5, L6: I don't think you have yet defined PBAP

We will define PBAP here.

P5, L12: sigma is not yet defined

We will explicitly state this as 3 standard deviations for clarity.

P6, L15: More clearly define what you mean by "identified" here.

We will revise this to:

> "A subset of approximately 17,000 particles were identified as exhibiting fluorescent intensities greater than the fluorescence threshold…"

P6, L25: "Significantly more aspherical" This relates to the comments by Referee #3 about AF.

> We will remove "significantly".

P8, L2: You combined clusters 2 and 4 into an assumed "PBAP" cluster, which is fine for analysis. The terminology applied here implies lesser uncertainty than I think is appropriate, however. I strongly suggest calling this something like "PBAP cluster" etc. so that it is easy to distinguish between "PBAP" as directly measured and "PBAP cluster" from those particles assigned to the name after clustering of fluorescent particles.

> We will revise this as suggested.

P12, L4: I agree with the authors that including this first-approximation of flux values is worthwhile here, but I think it is reasonable to heed the referee comments here by at least putting the magnitude of bounds on the uncertainty. I.e. are you talking about order of magnitude uncertainty? Plus/minus 20%, off by 3 order of magnitude, etc.

> We will revise the flux estimate section of the paper after revisiting referee #3's comments about their applicability and purpose they serve. This was meant as cautionary tale for previous flux estimates using this approach. We will include error bounds on the presented values using the standard deviations of the observed number concentrations.

P14, L20: Word "proven" is a bit strong here.

> We will remove "proven" in the revised manuscript.

P16, L1: ",based on …. comparison with … laboratory studies."

> We will revise this as suggested.

P16, L32: "Four typical pollens" … do you mean that they were all aerosolized together? Please clarify.

> The samples were all aerosolised separately. We will revise the previous sentence to the following to clarify this:

>> "Sample particles of interest were separately aerosolised into a large, clean HEPA filtered containment chamber (incorporating a recirculation fan), from which the WIBS-3D drew measurement samples. Dry materials were aerosolised directly from small quantities of powder using a filtered compressed air jet. The sample chamber was cleaned with absorbent paper and sodium hypochlorite in between samples to minimise contamination (Ruske et al., 2017)."

P17, Table A1: If Cl2 is likely to be pollen, what is Cl1? Please clarify

> Cluster Cl1 is unidentified but may be caused by fragmentation of the pollen samples during aerosolisation. We will include this after the discussion of Cl2 in the revised manuscript.

**Referee #3 Report**

The authors have clarified their selection of time periods to analyze, however their responses to other comments by myself and other reviewers are rather minimal and unsatisfying (i.e. I don't see where they have amended anything in response to reviewer 2's questions about the back trajectory analysis).

Additionally, now that the authors have included the number of particles observed in each cluster, I wonder if this might be a bit of over-analysis given the low statistics. Especially given that it seems like a single anthropogenic source from a ship could be responsible for quite a lot of what seemed to be PBAP. Even before this addition, multiple reviewers had commented about the robustness and applicability of these measurements and, in response, the authors have added one sentence at the end calling for more work in the future. In my opinion a larger and more thoughtful re-write would have been appropriate.

> We feel that the low counts of PBAP observed here serves to highlight the utility of the UV-LIF technique in environments where low numbers of particles of interest may be masked by several orders of magnitude more of common aerosol (e.g., sea salt) when using traditional collection methods, such as impaction on filters. We disagree that the other referees have called into question the applicability and robustness of the measurements in their assessment of the manuscript, however, we will revise our conclusions to include the possibility that the PBAP observed during wind event A may also be due to anthropogenic emissions from marine traffic. However, based on previous measurements of anthropogenic aerosol emissions this is unlikely
>
> We will revise P16L17 in the summary & conclusions to the following:
>
>> "The presence of particles characteristic of pollen is evidence towards the latter conclusion, however, the dispersion model results also display emissions from regions which feature significant marine traffic, and thus anthropogenic marine sources for these observations cannot be ruled out."
>
> During the review phase of this manuscript, the Savage et al., (2017) WIBS bioaerosol characterisation paper was accepted into public discussion. As part of this paper the authors investigated the effect of increasing the fluorescence threshold from the forced trigger mean + 3 standard deviations to the mean + 9 standard deviations in an attempt to minimise the inclusion of non-biological particles into the fluorescent population. They found that generally that all PBAP of interest would fluoresce well above this increased threshold, while non-biological interferents would not, effectively equating the fluorescent concentration to that of the true PBAP concentration. We have briefly investigated the effect of using this increased threshold in our analysis; this has the effect for greatly reducing the fluorescent number concentration to typically less than a few per litre, with peaks of 5 to 20 $L^{-1}$ during wind event A. The cluster analysis on this reduced input sample features 3 highly fluorescent clusters and 2 medium fluorescence clusters, relative to the new 9σ threshold. It was seen that using the increased threshold removes the likely non-biological material, leaving only

any significant PBAP concentrations during wind event A.  This is consistent with the results we present using the 3σ threshold.  The removal of weakly fluorescent non-biological aerosol from the clustering input improves the PBAP counting statistics to ~1000 particles observed during wind event A, as PBAP are not misattributed into the non-biological centroids as is possible in the 3σ threshold case.

I am still uncomfortable with the flux calculation as presented though I suppose the main conclusion is just that the local source, if there is one, is tiny. In response to my comment the authors say that they show that previous methods are overly simplistic but that's not at all how I read that section of the paper. To me it seems like the authors take these very simplistic tools and apply them to a scenario where it's not clear that they apply or give useful information but present it as the best that can be done.

The unconstrained, single height flux calculation presented is highly uncertain, as we state in the manuscript. After discussion of its scientific value, we chose to include the flux in the manuscript to highlight issues with this technique and raise questions about its applicability, since this method has been employed in frequently cited modelling studies.  Given the uncertainty of the method and the lack of other estimates to compare our values to, we feel it is best to revise the flux section to firmly state the uncertainty in the presented values and to advise caution in interpreting a result which may not be applicable to the circumstances of the measurements. The point here is that previous similar approaches have been used to estimate bioaerosol fluxes for model assimilation and such approaches can, as highlighted by the discussions, be problematic.  We revise this section to the following:

"In the past, short term wind driven enhancements in number concentrations have been used to infer the existence of local surface sources, e.g., Sesartic and Dallafior (2011), however, deriving an aerosol flux from single height concentration measurements can lead to highly uncertain results, Petelski and Piskozub (2006) & Pryor et al., (2008).  If we assume that the majority of the larger fluorescent particles (clusters Cl1 and PBAP cluster) are locally re-suspended then a net flux for these could be estimated using the approach of Sesartic and Dallafior (2011), resulting in fluxes for wind event A of $F_{Cl1}$ = 7.2 ± 11.8, and $F_{PBAP}$ = 1.1 ± 2.6 m$^2$s$^{-1}$.  However such calculations based on these crude assumptions are very uncertain.

Air mass back trajectory analysis and dispersion modelling (next section) reveal that the aerosol observed during wind event A may be from distal sources, and as such, the local surface source flux estimates presented are invalid in this case.  These values are presented to highlight the difficulty in estimating bioaerosol fluxes using these methods, which have previously been used in model assimilations, and they should not be over-interpreted given their uncertainty and may be invalid in this case."

I also think there could be much more thoughtful additions regarding the variable performance of the clustering algorithm and the impacts of that on the conclusions. For example, if the asymmetry factor cannot distinguish the "fine detail" between biological particles and dust (which should actually look very different from each other) then I think it is wrong to try to interpret small

variations in that measurement as indicative of different clusters. What happens if that piece of information is omitted?

In our prior response to question 3 by referee #3 we state that the quadrant detector is incapable of resolving fine detail. By this we meant that it cannot detect surface morphology as is possible with a dual CMOS array (as in the Ruske et al., 2017 study), however, differentiation between spheroidal, ellipsoidal and rod like particles is possible which can provide a useful piece information alongside other data when interpreting cluster results. Certainly in the case of clusters 1 & 3 we feel the difference in AF values is sufficiently large to be able to separate the two when also considering their fluorescence and size values, whereas the much more fluorescent clusters can clearly be distinguished based on their fluorescent intensities and size.

As requested we have removed AF from the clustering to demonstrate its effect. This tends to have the effect of promoting fragmentation of the clusters into subclusters based on our previous research into HCA algorithms, however, this detail was omitted from Crawford et al., (2015) for the sake of brevity. We see in this case that the removal of AF results in cluster fragmentation; the previous 4 cluster solution with AF included is fragmented into 12 clusters when AF is omitted from the clustering input, which we display visually below.

[Figure]

**Figure 1. Cluster solutions with AF included.**

[Figure]

**Figure 2. Cluster solutions with AF included.**

Here it can be seen that the original cluster 3 (Fig. 2) has been split into clusters 4 and 12 when AF is omitted (Fig. 3) on the basis of their size; original cluster 1 has been split into clusters 3 and 10 for the same reason; the remaining moderately to highly fluorescent fragmented clusters by in large compare with original PBAP clusters 2 & 4, the fragmentation of which is largely depended on the choice of HCA linkage used. Given the low numbers of particles assigned to each of these fragmented PBAP clusters (most contain single digit counts) we would still combine them into one larger overall PBAB cluster in our analysis.

In general, though the authors have at least responded to some of the larger comments, it still feels like the dataset is a bit oversold and the conclusions are simply not completely convincing or particularly interesting.

It was our intention to demonstrate the value of using UV-LIF spectrometers to detect very low counts of PBAP in a climate sensitive region, which may be missed by traditional methods, rather than to present the results as being generally representative of Antarctic bioaerosol. We feel that this dataset shows that PBAP may be transported from South America, as hypothesised by Pearce et al., (2009), and is of interest to the community for this reason. We accept that other sources such as marine traffic may also be the source of the observed PBAP and this will be clarified in the revised manuscript as detailed in an earlier response. We are encouraged however that the observations have generated new discussions.

[revised manuscript text omitted]